# Embracing the chaos: analysis and diagnosis of numerical instability in variational flows

**Zuheng Xu**     **Trevor Campbell**
Department of Statistics
University of British Columbia
[zuheng.xu | trevor]@stat.ubc.ca

## Abstract

In this paper, we investigate the impact of numerical instability on the reliability of sampling, density evaluation, and evidence lower bound (ELBO) estimation in variational flows. We first empirically demonstrate that common flows can exhibit a catastrophic accumulation of error: the numerical flow map deviates significantly from the exact map—which affects sampling—and the numerical inverse flow map does not accurately recover the initial input—which affects density and ELBO computations. Surprisingly though, we find that results produced by flows are often accurate enough for applications despite the presence of serious numerical instability. In this work, we treat variational flows as dynamical systems, and leverage *shadowing theory* to elucidate this behavior via theoretical guarantees on the error of sampling, density evaluation, and ELBO estimation. Finally, we develop and empirically test a diagnostic procedure that can be used to validate results produced by numerically unstable flows in practice.

## 1 Introduction

Variational families of probability distributions play a prominent role in generative modelling and probabilistic inference. A standard construction of a variational family involves passing a simple reference distribution—such as a standard normal—through a sequence of parametrized invertible transformations, i.e., a *flow* [1–7]. Modern flow-based variational families have the representational capacity to adapt to highly complex target distributions while still providing computationally tractable i.i.d. draws and density evaluations, which in turn enables convenient training by minimizing the Kullback-Leibler (KL) divergence [8] via stochastic optimization methods [9–11].

In practice, creating a flexible variational flow often requires many flow layers, as the expressive power of a variational flow generally increases with the number of transformations [5, 12, 13]. Composing many transformations in this manner, however, can create numerical instability [5, 14–16], i.e., the tendency for numerical errors from imprecise floating-point computations to be quickly magnified along the flow. This accumulation of error can result in low-quality sample generation [14], numerical non-invertibility [14, 15], and unstable training [16]. But in this work, we demonstrate that this is not always the case with a counterintuitive empirical example: a flow that exhibits severe numerical instability—with error that grows exponentially in the length of the flow—but which surprisingly still returns accurate density values, i.i.d. draws, and evidence lower bound (ELBO) estimates.

Motivated by this example, the goal of this work is twofold: (1) to provide a theoretical understanding of how numerical instability in flows relates to error in downstream results; and (2) to provide a diagnostic procedure such that users of normalizing flows can check whether their particular results are reliable in practice. We develop a theoretical framework that investigates the influence of numerical instability using *shadowing theory* from dynamical systems [17–20]. Intuitively, shadowing theory asserts that although a numerical flow trajectory may diverge quickly from its exact counterpart,

37th Conference on Neural Information Processing Systems (NeurIPS 2023).

there often exists another exact trajectory nearby that remains close to the numerical trajectory (the *shadowing property*). We provide theoretical error bounds for three core tasks given a flow exhibiting the shadowing property—sample generation, density evaluation, and ELBO estimation—that show that the error grows much more slowly than the error of the trajectory itself. Our results pertain to both normalizing flows [2, 3] and mixed variational flows [5]. We develop a diagnostic procedure for practical verification of the shadowing property of a given flow. Finally, we validate our theory and diagnostic procedure on MixFlow [5] on both synthetic and real data examples, and Hamiltonian variational flow [6] on synthetic examples.

**Related work.** Chang et al. [21] connected the numerical invertibility of deep residual networks (ResNets) and ODE stability analysis by interpreting ResNets as discretized differential equations, but did not provide a quantitative analysis of how stability affects the error of sample generation or density evaluation. This approach also does not apply to general flow transformations such as coupling layers. Behrmann et al. [14] analyzed the numerical inversion error of generic normalizing flows via bi-Lipschitz continuity for each flow layer, resulting in error bounds that grow exponentially with flow length. These bounds do not reflect empirical results in which the error in downstream statistical procedures (sampling, density evaluation, and ELBO estimation) tends to grow much more slowly. Beyond the variational literature, Bahsoun et al. [22] investigated statistical properties of numerical trajectories by modeling numerical round-off errors as small random perturbations, thus viewing the numerical trajectory as a sample path of a Markov chain. However, their analysis was constrained by the requirement that the exact trajectory follows a time-homogeneous, strongly mixing measure-preserving dynamical system, a condition that does not generally hold for variational flows. Tupper [23] established a relationship between shadowing and the weak distance between inexact and exact orbits in dynamical systems. Unlike our work, their results are limited to quantifying the quality of pushforward samples and do not extend to the evaluation of densities and ELBO estimation. In contrast, our error analysis provides a more comprehensive understanding of the numerical error of all three downstream statistical procedures.

## 2 Background: variational flows

A *variational family* is a parametrized set of probability distributions $\{q_\lambda : \lambda \in \Lambda\}$ on some space $\mathcal{X}$ that each enable tractable i.i.d. sampling and density evaluation. The ability to obtain draws and compute density values is crucial for fitting the family to data via maximum likelihood in generative modelling, as well as maximizing the ELBO in variational inference,

$$\text{ELBO}\,(q_\lambda||p) = \int q_\lambda(x) \log \frac{p(x)}{q_\lambda(x)} \mathrm{d}x,$$

where $p(x)$ is an unnormalized density corresponding to a target probability distribution $\pi$. In this work we focus on $\mathcal{X} \subseteq \mathbb{R}^d$ endowed with its Borel $\sigma$-algebra and Euclidean norm $\|\cdot\|$, and parameter set $\Lambda \subseteq \mathbb{R}^p$. We assume all distributions have densities with respect to a common base measure on $\mathcal{X}$, and will use the same symbol to denote a distribution and its density. Finally, we will suppress the parameter subscript $\lambda$, since we consider a fixed member of a variational family throughout.

**Normalizing flows.** One approach to building a variational distribution $q$ is to push a reference distribution, $q_0$, through a measurable bijection on $\mathcal{X}$. To ensure the function is flexible yet still enables tractable computation, it is often constructed via the composition of simpler measurable, invertible "layers" $F_1, \ldots, F_N$, referred together as a (normalizing) *flow* [1, 2, 4]. To generate a draw $Y \sim q$, we draw $X \sim q_0$ and evaluate $Y = F_N \circ F_{N-1} \circ \cdots \circ F_1(X)$. When each $F_n$ is a diffeomorphism,[1] the density of $q$ is

$$\forall x \in \mathcal{X} \qquad q(x) = \frac{q_0(F_1^{-1} \circ \cdots \circ F_N^{-1}(x))}{\prod_{n=1}^N J_n(F_n^{-1} \circ \cdots \circ F_N^{-1}(x))} \qquad J_n(x) = |\det \nabla_x F_n(x)|. \quad (1)$$

An unbiased ELBO estimate can be obtained using a draw from $q_0$ via

$$X \sim q_0, \ E(X) = \log p(F_N \circ \cdots \circ F_1(X)) - \log q_0(X) + \sum_{n=1}^N \log J_n(F_n \circ \cdots \circ F_1(X)). \quad (2)$$

---

[1]A differentiable map with a differentiable inverse; we assume all flow maps in this work are diffeomorphisms.

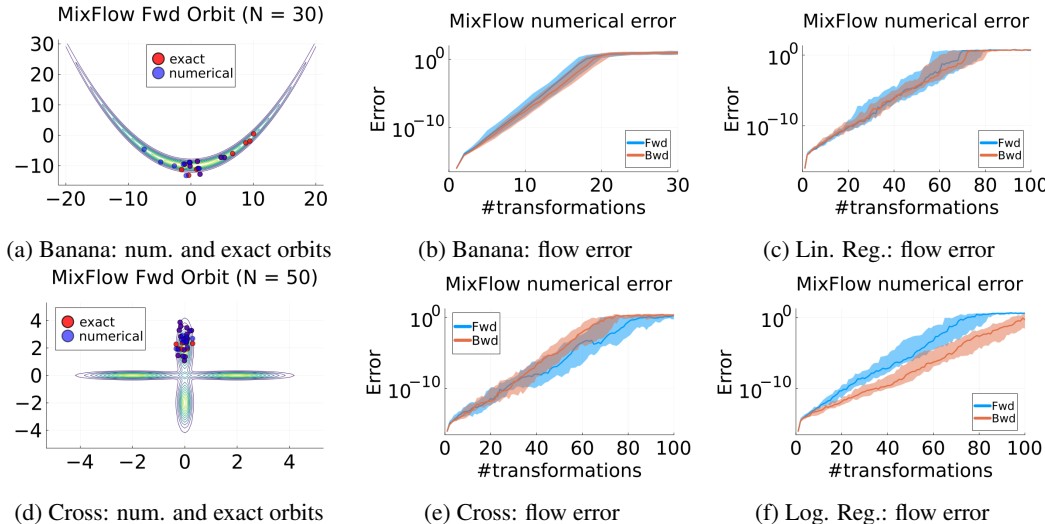

(a) Banana: num. and exact orbits     (b) Banana: flow error     (c) Lin. Reg.: flow error

(d) Cross: num. and exact orbits     (e) Cross: flow error     (f) Log. Reg.: flow error

Figure 1: MixFlow forward (`fwd`) and backward (`bwd`) orbit errors on the Banana and Cross distributions, and on two real data examples. The first column visualizes the `exact` and `numerical` orbits with the same starting point on the synthetic targets. For a better visualization, we display every 2nd and 4th states of presented orbits (instead of the complete orbits) in Figs. 1a and 1d, respectively. Figs. 1b, 1c, 1e and 1f show the median and upper/lower quartile forward error $\|F^k x - \hat{F}^k x\|$ and backward error $\|B^k x - \hat{B}^k x\|$ comparing $k$ transformations of the forward exact/approximate maps $F$, $\hat{F}$ or backward exact/approximate maps $B$, $\hat{B}$. Statistics are plotted over 100 initialization draws from the reference distribution $q_0$. For the `exact` maps we use a 2048-bit `BigFloat` representation, and for the `numerical` approximations we use 64-bit `Float` representation. The "exactness" of `BigFloat` representation is justified in Fig. 15 in Appendix D.3.

Common choices of $F_n$ are RealNVP [24], neural spline [25], Sylvester [26], Hamiltonian-based [6, 7], and planar and radial [2] flows; see Papamakarios et al. [3] for a comprehensive overview.

**MixFlows.** Instead of using only the final pushforward distribution as the variational distribution $q$, Xu et al. [5] proposed averaging over all the pushforwards along a flow trajectory with identical flow layers, i.e., $F_n = F$ for some fixed $F$. When $F$ is ergodic and measure-preserving for some target distribution $\pi$, Xu et al. [5] show that averaged flows guarantee total variation convergence to $\pi$ in the limit of increasing flow length. To generate a draw $Y \sim q$, we first draw $X \sim q_0$ and a flow length $K \sim \mathsf{Unif}\{0, 1, \ldots, N\}$, and then evaluate $Y = F^K(X) = F \circ F \circ \cdots \circ F(X)$ via $K$ iterations of the map $F$. The density formula is given by the mixture of intermediate flow densities,

$$\forall x \in \mathcal{X}, \qquad q(x) = \frac{1}{N+1} \sum_{n=0}^{N} \frac{q_0(F^{-n}x)}{\prod_{j=1}^{n} J(F^{-j}x)}, \quad J(x) = |\det \nabla_x F(x)|. \tag{3}$$

An unbiased ELBO estimate can be obtained using a draw from $q_0$ via

$$X \sim q_0, \quad E(X) = \frac{1}{N+1} \sum_{n=0}^{N} \log \frac{p(F^n X)}{q(F^n X)}. \tag{4}$$

## 3   Instability in variational flows and layer-wise error analysis

In practice, normalizing flows often involve tens of flow layers, with each layer coupled with deep neural network blocks [2, 4, 24] or discretized ODE simulations [6, 7]; past work on MixFlows requires thousands of flow transformations [5]. These flows are typically chaotic and sensitive to small perturbations, such as floating-point representation errors. As a result, the final output of a numerically computed flow can significantly deviate from its exact counterpart, which may cause downstream issues during training or evaluation of the flow [12, 14, 27]. To understand the accumulation of error,

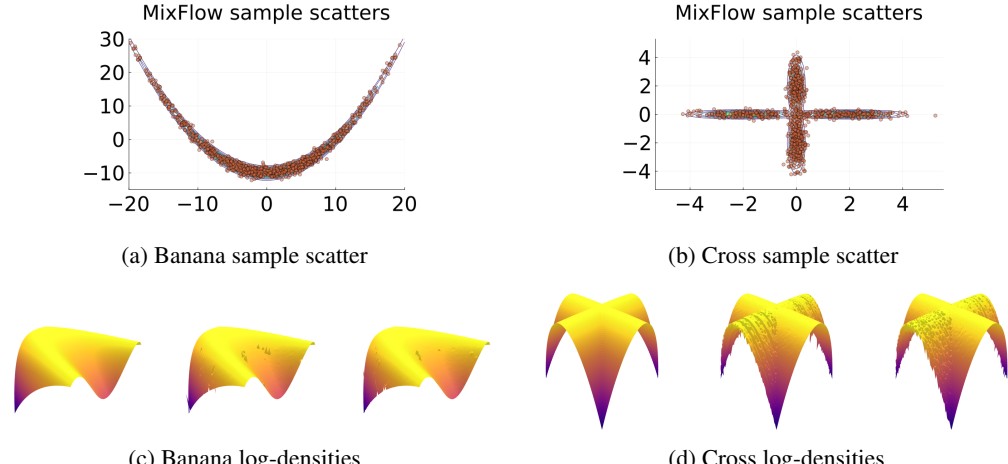

(a) Banana sample scatter

(b) Cross sample scatter

(c) Banana log-densities

(d) Cross log-densities

Figure 2: Figs. 2a and 2b respectively show sample scatters produced by the naïve application of MixFlow formulae targeting at the banana and cross distributions, without accounting for numerical error. Figs. 2c and 2d display comparisons of log-densities on both synthetic examples. The true log-target is on the left, the exact MixFlow evaluation (computed via 2048-bit `BigFloat` representation) is in the middle, and the numerical MixFlow evaluation is on the right.

Behrmann et al. [14] assume that each flow layer $F_n$ and its inverse are Lipschitz and have bounded error, $\sup_x \|F_n(x) - \hat{F}_n(x)\| \leq \delta$, and analyze the error of each layer individually. This layer-wise analysis tends to yield error bounds that grow exponentially in the flow length $N$ (see Appendix A.1):

$$\|F_N \circ \cdots \circ F_1(x) - \hat{F}_N \circ \cdots \circ \hat{F}_1(x)\| \leq \delta \sum_{n=0}^{N-1} \prod_{j=n+2}^{N} \mathrm{Lip}(F_j). \tag{5}$$

Given a constant flow map $F_j = F$ with Lipschitz constant $\ell$, the bound is $O(\delta \ell^N)$, which suggests that the error accumulates exponentially in the length of the flow. In Fig. 1, we test this hypothesis empirically using MixFlows on two synthetic targets (the banana and cross distribution) and two real data examples (Bayesian linear regression and logistic problems) taken from past work on MixFlows [5]. See Appendix D for the details of this test. Figs. 1b, 1c, 1e and 1f confirms that the exponential growth in the error bound Eq. (5) is reasonable; the error does indeed grow exponentially quickly in practice. And Fig. 16 (a)–(b) in Appendix D.3 further demonstrate that after fewer than 100 transformations both flows have error on the same order of magnitude as the scale of the target distribution. Naïvely, this implies that sampling, density evaluation, and ELBO estimation may be corrupted badly by numerical error.

But counterintuitively, in Fig. 2 we find that simply ignoring the issue of numerical error and using the exact formulae yield reasonable-looking density evaluations and sample scatters. These results are of course only qualitative in nature; we provide corresponding quantitative results later in Section 6. But Fig. 2 appears to violate the principle of "garbage in, garbage out;" the buildup of significant numerical error in the sample trajectories themselves does not seem to have a strong effect on the quality of downstream sampling and density evaluation. The remainder of this paper is dedicated to resolving this counterintuitive behavior using shadowing theory [28].

## 4 Global error analysis of variational flows via shadowing

### 4.1 The shadowing property

We analyze variational flows as finite discrete-time dynamical systems (see, e.g., [28]). In this work, we consider a *dynamical system* to be a sequence of diffeomorphisms on $(\mathcal{X}, \|\cdot\|)$. We define the *forward dynamics* $(F_n)_{n=1}^{N}$ to be the flow layer sequence, and the *backward dynamics* $(B_n)_{n=1}^{N}$ to be comprised of the inverted flow layers $B_n = F_{N-(n-1)}^{-1}$ in reverse order. An *orbit* of a dynamical system starting at $x \in \mathcal{X}$ is the sequence of states produced by the sequence of maps when initialized

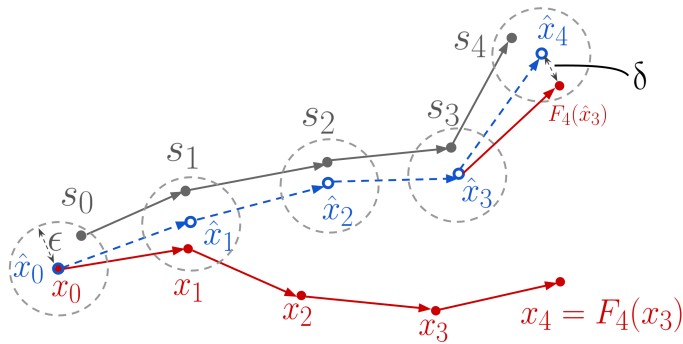

Figure 3: A visualization of a pseudo-orbit and shadowing orbit. Solid arrows and filled dots indicate exact orbits, while dashed arrows and open dots indicate pseudo-orbits (e.g., via numerical computations). Red indicates the exact orbit $(x_0, \ldots, x_4)$ that one intends to compute. Blue indicates the numerically computed $\delta$-pseudo-orbit $(\hat{x}_0, \ldots, \hat{x}_4)$. Grey indicates the corresponding $\epsilon$-shadowing orbit $(s_1, \ldots, s_4)$. At the top right of the figure, $\|\hat{x}_4 - F_4(\hat{x}_3)\| \leq \delta$ illustrates the $\delta$ numerical error at each step of the pseudo-orbit. The grey dashed circles demonstrate the $\epsilon$-shadowing window.

at $x$. Therefore, the forward and backward orbits initialized at $x \in \mathcal{X}$ are

$$\text{Forward Orbit:} \quad x = x_0 \xrightarrow{F_1} x_1 \xrightarrow{F_2} x_2 \to \ldots \xrightarrow{F_N} x_N$$

$$x_{-N} \xleftarrow{B_N = F_1^{-1}} \cdots \leftarrow x_{-2} \xleftarrow{B_2 = F_{N-1}^{-1}} x_{-1} \xleftarrow{B_1 = F_N^{-1}} x_0 = x \quad : \text{Backward Orbit.}$$

Given numerical implementations $\hat{F}_n \approx F_n$ and $\hat{B}_n \approx B_n$ with tolerance $\delta > 0$, i.e.,

$$\forall x \in \mathcal{X}, \quad \|F_n(x) - \hat{F}_n(x)\| \leq \delta, \quad \|B_n(x) - \hat{B}_n(x)\| \leq \delta, \tag{6}$$

we define the *forward* and *backward pseudo-dynamics* to be $(\hat{F}_n)_{n=1}^N$ and $(\hat{B}_n)_{n=1}^N$, respectively, along with their *forward* and *backward pseudo-orbits* initialized at $x \in \mathcal{X}$:

$$\text{Forward Pseudo-Orbit:} \quad x = \hat{x}_0 \xrightarrow{\hat{F}_1 \approx F_1} \hat{x}_1 \xrightarrow{\hat{F}_2 \approx F_2} \hat{x}_2 \to \ldots \xrightarrow{\hat{F}_N \approx F_N} \hat{x}_N$$

$$\hat{x}_{-N} \xleftarrow{\hat{B}_N \approx F_1^{-1}} \cdots \leftarrow \hat{x}_{-2} \xleftarrow{\hat{B}_2 \approx F_{N-1}^{-1}} \hat{x}_{-1} \xleftarrow{\hat{B}_1 \approx F_N^{-1}} \hat{x}_0 = x \quad : \text{Backward Pseudo-Orbit.}$$

For notational brevity, we use subscripts on elements of $\mathcal{X}$ throughout to denote forward/backward orbit states. For example, given a random element $Z \in \mathcal{X}$ drawn from some distribution, $Z_k$ is the $k^{\text{th}}$ state generated by the forward dynamics $(F_n)_{n=1}^N$ initialized at $Z$, and $Z_{-k}$ is the $k^{\text{th}}$ state generated by the backward dynamics $(B_n)_{n=1}^N$ initialized at $Z$. Hat accents denote the same for the pseudo-dynamics: for example, $\hat{Z}_k$ is the $k^{\text{th}}$ state generated by $(\hat{F}_n)_{n=1}^N$ when initialized at $Z$.

The forward/backward orbits satisfy $x_{k+1} = F_k(x_k)$ and $x_{-(k+1)} = B_k(x_{-k})$ exactly at each step. On the other hand, the forward/backward pseudo-orbits incur a small amount of error,

$$\|\hat{x}_{k+1} - F_k(\hat{x}_k)\| \leq \delta \quad \text{and} \quad \|\hat{x}_{-(k+1)} - B_k(\hat{x}_{-k})\| \leq \delta,$$

which can be magnified quickly along the orbit. However, there often exists another *exact* orbit starting at some other point $s \in \mathcal{X}$ that remains in a close neighbourhood of the numerical orbit. This property, illustrated in Fig. 3, is referred to as the *shadowing property* [19, 28–31].

**Definition 4.1** (Shadowing property [28])**.** *The forward dynamics* $(F_n)_{n=1}^N$ *has the* **$(\epsilon, \delta)$-shadowing property** *if for all* $x \in \mathcal{X}$ *and all* $(\hat{F}_n)_{n=1}^N$ *satisfying Eq. (6), there exists an* $s \in \mathcal{X}$ *such that*

$$\forall k = 0, 1, \ldots, N, \quad \|s_k - \hat{x}_k\| < \epsilon.$$

An analogous definition holds for the backward dynamics $(B_n)_{n=1}^N$—where there is a shadowing orbit $s_{-k}$ that is nearby the pseudo-orbit $\hat{x}_{-k}$—and for the joint forward and backward dynamics, where there is a shadowing orbit nearby both the backward and forward pseudo-orbits simultaneously. The key idea in this paper is that, intuitively, if the numerically computed pseudo-orbit is close to

*some* exact orbit (the shadowing orbit), statistical computations based on the pseudo-orbit—e.g., sampling, density evaluation, and ELBO estimation—should be close to those obtained via that exact orbit. We will defer the examination of when (and to what extent) shadowing holds to Section 5; in this section, we will use it as an assumption when analyzing statistical computations with numerically implemented variational flows.

## 4.2 Error analysis of normalizing flows via shadowing

**Sampling.** Our first goal is to relate the marginal distributions of $X_N$ and $\hat{X}_N$ for $X \sim q_0$, i.e., to quantify the error in sampling due to numerical approximation. Assume the normalizing flow has the $(\epsilon, \delta)$-shadowing property, and let $\xi_0$ be the marginal distribution of the shadowing orbit start point. We suspect that $\xi_0 \approx q_0$, in some sense; for example, we know that the bounded Lipschitz distance $D_{BL}(\xi_0, q_0)$ is at most $\epsilon$ due to shadowing. And $\xi_0$ is indeed an implicit function of $q_0$; it is a fixed point of a twice differentiable function involving the whole orbit starting at $q_0$ [17, Page. 176]. But the distribution $\xi_0$ is generally hard to describe more completely, and thus it is common to impose additional assumptions. For example, past work shows that the Lévy-Prokhorov metric $D_{LP}(\hat{X}_N, X_N)$ is bounded by $\epsilon$, under the assumption that $\xi_0 = q_0$ [23]. We provide a more general result (Proposition 4.2) without distributional assumptions on $\xi_0$. We control $D_{BL}(\cdot, \cdot)$ rather than $D_{LP}(\cdot, \cdot)$, as its analysis is simpler and both metrize weak distance.

**Proposition 4.2.** *Suppose the forward dynamics has the $(\epsilon, \delta)$-shadowing property, and $X \sim q_0$. Then*

$$\sup_{f:|f|\leq U, \mathrm{Lip}(f)\leq \ell} \left| \mathbb{E}f(X_N) - \mathbb{E}f(\hat{X}_N) \right| \leq \ell\epsilon + 2U D_{TV}(\xi_0, q_0).$$

*In particular, with $\ell = U = 1$, we obtain that $D_{BL}(X_N, \hat{X}_N) \leq \epsilon + 2D_{TV}(\xi_0, q_0)$.*

Recall the layerwise error bound from Eq. (5)—which suggests that the difference in orbit and pseudo-orbit grows exponentially in $N$—and compare to Proposition 4.2, which asserts that the error is controlled by the shadowing window size $\epsilon$. This window size may depend on $N$, but we find in Section 6 it is usually not much larger than $\delta$ in practice, which itself is typically near the precision of the relevant numerical representation. We will show how to estimate $\epsilon$ later in Section 5.

**Density evaluation.** The exact density $q(x)$ follows Eq. (1), while the approximation $\hat{q}(x)$ is the same except that we use the backward *pseudo*-dynamics. For $x \in \mathcal{X}$, a differentiable function $g : \mathcal{X} \mapsto \mathbb{R}^+$, define the *local Lipschitz constant* for the logarithm of $g$ around $x$ as:

$$L_{g,\epsilon}(x) = \sup_{\|y-x\|\leq\epsilon} \|\nabla \log g(y)\|.$$

Theorem 4.3 shows that, given the shadowing property, the numerical error is controlled by the shadowing window size $\epsilon$ and the sum of the Lipschitz constants along the pseudo-orbit. The constant $L_{q,\epsilon}$ occurs because we are essentially evaluating $q(s)$ rather than $q(x)$, where $s \in \mathcal{X}$ is the backward shadowing orbit initialization. The remaining constants occur because of the approximation of the shadowing orbit with the nearby, numerically-computed pseudo-orbit in the density formula.

**Theorem 4.3.** *Suppose the backward dynamics has the $(\epsilon, \delta)$-shadowing property. Then*

$$\forall x \in \mathcal{X}, \quad |\log \hat{q}(x) - \log q(x)| \leq \epsilon \cdot \left( L_{q,\epsilon}(x) + L_{q_0,\epsilon}(\hat{x}_{-N}) + \sum_{n=1}^{N} L_{J_{N-n+1,\epsilon}}(\hat{x}_{-n}) \right).$$

**ELBO estimation.** The exact ELBO estimation function is given in Eq. (2); the numerical ELBO estimation function $\hat{E}(x)$ is the same except that we use the forward *pseudo*-dynamics. The quantity $E(X), X \sim q_0$ is an unbiased estimate of the exact ELBO, while $\hat{E}(X)$ is biased by numerical error; Theorem 4.4 quantifies this error. Note that for simplicity we assume that the initial state distributions $q_0, \xi_0$ described earlier are identical. It is possible to obtain a bound including a $D_{TV}(q_0, \xi_0)$ term, but this would require the assumption that $\log p(x)/q(x)$ is uniformly bounded on $\mathcal{X}$.

**Theorem 4.4.** *Suppose the forward dynamics has the $(\epsilon, \delta)$-shadowing property, and $\xi_0 = q_0$. Then*

$$\left| \mathrm{ELBO}\,(q\|p) - \mathbb{E}[\hat{E}(X)] \right| \leq \epsilon \cdot \mathbb{E}\left[ L_{p,\epsilon}(\hat{X}_N) + L_{q_0,\epsilon}(X) + \sum_{n=1}^{N} L_{J_n,\epsilon}(\hat{X}_n) \right] \qquad \textit{for } X \sim q_0.$$

### 4.3 Error analysis of MixFlows via shadowing

The error analysis for MixFlows for finite length $N \in \mathbb{N}$ parallels that of normalizing flows, except that $F_n = F$, $B_n = B$, and $J_n = J$ for $n = 1, \ldots, N$. However, when $F$ and $B$ are ergodic and measure-preserving for the target $\pi$, we provide asymptotically simpler bounds in the large $N$ limit that do not depend on the difficult-to-analyze details of the Lipschitz constants along a pseudo-orbit. These results show that the error of sampling tends to be constant in flow length, while the error of density evaluation and ELBO estimation grows at most linearly. We say the forward dynamics has the *infinite $(\epsilon, \delta)$-shadowing property* if $(F_n)_{n=1}^{\infty}$ has the $(\epsilon, \delta)$-shadowing property [32]. Analogous definitions hold for both the backward and joint forward/backward dynamics.

**Sampling.** Similar to Proposition 4.2 for normalizing flows, Proposition 4.5 bounds the error in exact draws $Y$ and approximate draws $\hat{Y}$ from the MixFlow. The result demonstrates that error does not directly depend on the flow length $N$, but rather on the shadowing window $\epsilon$. In addition, in the setting where the flow map $F$ is $\pi$-ergodic and measure-preserving, we can (asymptotically) remove the total variation term between the initial shadowing distribution $\xi_0$ and the reference distribution $q_0$. Note that in the asymptotic bound, the distributions of $Y$ and $\hat{Y}$ are functions of $N$.

**Proposition 4.5.** *Suppose the forward dynamics has the $(\epsilon, \delta)$-shadowing property, and $X \sim q_0$. Let $Y = X_K$, $\hat{Y} = \hat{X}_K$ for $K \sim \mathsf{Unif}\{0, 1, \ldots, N\}$. Then*

$$\sup_{f : |f| \leq U, \mathrm{Lip}(f) \leq \ell} \left| \mathbb{E}f(Y) - \mathbb{E}f(\hat{Y}) \right| \leq \ell\epsilon + 2U\mathrm{D}_{\mathrm{TV}}\left(\xi_0, q_0\right).$$

*In particular, if $\ell = U = 1$, we obtain that $\mathrm{D}_{\mathrm{BL}}(Y, \hat{Y}) \leq \epsilon + 2\mathrm{D}_{\mathrm{TV}}(\xi_0, q_0)$. If additionally the forward dynamics has the infinite $(\epsilon, \delta)$-shadowing property, $F$ is $\pi$-ergodic and measure-preserving, and $\xi_0, q_0 \ll \pi$, then*

$$\limsup_{N \to \infty} \mathrm{D}_{\mathrm{BL}}\left(Y, \hat{Y}\right) \leq \epsilon.$$

A direct corollary of the second result in Proposition 4.5 is that for $W \sim \pi$, $\lim_{N \to \infty} \mathrm{D}_{\mathrm{BL}}(W, \hat{Y}) \leq \epsilon$, which results from the fact that $\mathrm{D}_{\mathrm{BL}}(W, Y) \to 0$ [5, Theorem 4.1]. In other words, the bounded Lipschitz distance of the approximated MixFlow and the target $\pi$ is asymptotically controlled by the shadowing window $\epsilon$. This improves upon earlier theory governing approximate MixFlows, for which the best guarantee available had total variation error growing as $O(N)$ [5, Theorem 4.3].

**Density evaluation.** The exact density follows Eq. (3); the numerical approximation is the same except that we use the backward *pseudo*-dynamics. We obtain a similar finite-$N$ result as in Theorem 4.3. Further, we show that given infinite shadowing—where the shadowing window size $\epsilon$ is independent of flow length $N$—the numerical approximation error asymptotically grows at most linearly in $N$, in proportion to $\epsilon$.

**Theorem 4.6.** *Suppose the $(\epsilon, \delta)$-shadowing property holds for the backward dynamics. Then*

$$\forall x \in \mathcal{X}, \quad |\log \hat{q}(x) - \log q(x)| \leq \epsilon \cdot \left( L_{q,\epsilon}(x) + \max_{0 \leq n \leq N} L_{q_0,\epsilon}(\hat{x}_{-n}) + \sum_{n=1}^{N} L_{J,\epsilon}(\hat{x}_{-n}) \right).$$

*If additionally the backward dynamics has the infinite $(\epsilon, \delta)$-shadowing property, $F$ is $\pi$-ergodic and measure-preserving, $L_{q,\epsilon}(x) = o(N)$ as $N \to \infty$, and $\xi_0 \ll \pi$, then for $q_0$-almost every $x \in \mathcal{X}$,*

$$\limsup_{N \to \infty} \frac{1}{N} |\log \hat{q}(x) - \log q(x)| \leq \epsilon \cdot \mathbb{E}\left[L_{q_0,\epsilon}(X) + L_{J,\epsilon}(X)\right], \quad X \sim \pi.$$

**ELBO estimation.** The exact ELBO formula for the MixFlow is given in Eq. (4). Note that here we do not simply substitute the forward/backward pseudo-orbits as needed; the naïve approximation of the terms $q(F^n x)$ would involve $n$ applications of $\hat{F}$ followed by $N$ applications of $\hat{B}$, which do not exactly invert one another. Instead, we analyze the method proposed in [5], which involves simulating a single forward pseudo-orbit $\hat{x}_1, \ldots, \hat{x}_N$ and backward pseudo-orbit $\hat{x}_{-1}, \ldots, \hat{x}_{-N}$ starting from $x \in \mathcal{X}$, and then caching these and using them as needed in the exact formula.

**Theorem 4.7.** *Suppose the joint forward and backward dynamics has the $(\epsilon, \delta)$-shadowing property, and $\xi_0 = q_0$. Then for $X \sim q_0$,*

$$\left| \text{ELBO} \left( q \| p \right) - \mathbb{E} \left[ \hat{E}(X) \right] \right| \leq \epsilon \cdot \mathbb{E} \left[ \frac{1}{N+1} \sum_{n=0}^{N} L_{p,\epsilon}(\hat{X}_n) + \max_{-N \leq n \leq N} L_{q_0,\epsilon}(\hat{X}_n) + \sum_{n=-N}^{N} L_{J,\epsilon}(\hat{X}_n) \right].$$

*If additionally the joint forward and backward dynamics has the infinite $(\epsilon, \delta)$-shadowing property, $F$ is $\pi$-ergodic and measure-preserving, and for some $1 \leq m_1 < \infty$ and $1/m_1 + 1/m_2 = 1$ $L_{p,\epsilon}, L_{q_0,\epsilon}, L_{J,\epsilon} \in L^{m_1}(\pi)$ and $\frac{\mathrm{d}q_0}{\mathrm{d}\pi} \in L^{m_2}(\pi)$, then*

$$\limsup_{N \to \infty} \frac{1}{N} \left| \text{ELBO} \left( q \| p \right) - \mathbb{E} \left[ \hat{E}(X) \right] \right| \leq 2\epsilon \cdot \mathbb{E} \left[ L_{q_0,\epsilon}(X) + L_{J,\epsilon}(X) \right], \quad X \sim \pi.$$

## 5 Computation of the shadowing window size

We have so far assumed the $(\epsilon, \delta)$-shadowing property throughout. To make use of our results in practice, it is crucial to understand the shadowing window size $\epsilon$. Theorem 5.1 presents sufficient conditions for a finite dynamical system to have the shadowing property, and characterizes the size of the shadowing window $\epsilon$. Note that throughout this section we focus on the forward dynamics; the backward and joint dynamics can be treated identically. Let $\| \cdot \|$ denote the spectral norm of a matrix.

**Theorem 5.1** (Finite shadowing theorem). *Suppose the dynamics $(F_n)_{n=1}^{N}$ are $C^2$ diffeomorphisms on $\mathcal{X}$, and the pseudo-dynamics $(\hat{F}_n)_{n=1}^{N}$ satisfy Eq. (6). For a given $x \in \mathcal{X}$, define the operator $A : \mathcal{X}^{N+1} \mapsto \mathcal{X}^N$ by*

$$(Au)_k = u_{k+1} - \nabla F_{k+1}(\hat{x}_k)u_k, \quad \text{for } u = (u_0, u_1, \dots, u_N) \in \mathcal{X}^N, \quad k = 0, 1, \dots, N-1.$$

*Let*

$$M := \max \left\{ \sup_{\|v\| \leq 2\lambda\delta} \|\nabla^2 F_{n+1}(\hat{x}_n + v)\| : n = 0, 1, \dots, N-1 \right\} \text{ where } \lambda = \lambda_{min}(AA^T)^{-1/2}.$$

*If $2M\lambda^2\delta \leq 1$, then the pseudo-orbit starting at $x$ is shadowed with $\epsilon = 2\lambda\delta$.*

*Proof.* This result follows directly by following the proof of [18, Theorem 11.3] with nonconstant flow maps $F_n$, and then using the right-inverse norm from [33, Corollary 4.2]. $\square$

In order to apply Theorem 5.1 we need to (1) estimate $\delta$, e.g., by examining numerical error of one step of the map in practice; (2) compute $\lambda_{\min}(AA^T)$; and (3) estimate $M$, e.g., by bounding the third derivative of $F_n$. While estimating $M$ and $\delta$ are problem-specific, one can employ standard procedures for computing $\lambda_{\min}(AA^T)$. The matrix $AA^T$ has a block-tridiagonal form,

$$AA^T = \begin{bmatrix} D_1 D_1^T + I & -D_2^T & & & & \\ -D_2 & D_2 D_2^T + I & -D_3^T & & & \\ & \ddots & \ddots & \ddots & & \\ & & -D_{N-1} & D_{N-1} D_{N-1}^T + I & -D_N^T \\ & & & -D_N & D_N D_N^T + I \end{bmatrix},$$

where $D_k = \nabla F_k(\hat{x}_{k-1}) \in \mathbb{R}^{d \times d}, \forall k \in [N]$. Notice that $AA^T$ is a symmetric positive definite sparse matrix with bandwidth $d$ and so has $O(Nd^2)$ entries. The inherent structured sparsity can be leveraged to design efficient eigenvalue computation methods, e.g., the inverse power method [34], or tridiagonalization via Lanczos iterations [35] followed by divide-and-conquer algorithms [36]. However, in our experiments, directly calling the `eigmin` function provided in Julia suffices; as illustrated in Figs. 14 and 19, the shadowing window computation requires only a few seconds for low dimensional synthetic examples with hundreds flow layers, and several minutes for real data examples. Hence, we didn't pursue specialized methods, leaving that for future work. It is also noteworthy that practical computations of $D_k$ can introduce floating-point errors, influencing the accuracy of $\lambda$. To address this, one might consider adjusting the shadowing window size. We explain how to manage this in Appendix B.1, and why such numerical discrepancies minimally impact results.

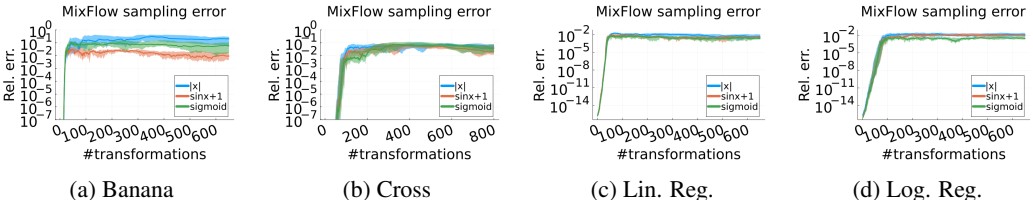

(a) Banana     (b) Cross     (c) Lin. Reg.     (d) Log. Reg.

Figure 4: MixFlow relative sample average computation error on three test functions: $\sum_{i=1}^{d}[|x|]_i, \sum_{i=1}^{d}[\sin(x)+1]_i$ and $\sum_{i=1}^{d}[\mathrm{sigmoid}(x)]_i$. The lines indicate the median, and error regions indicate $25^{\mathrm{th}}$ to $75^{\mathrm{th}}$ percentile from 20 runs.

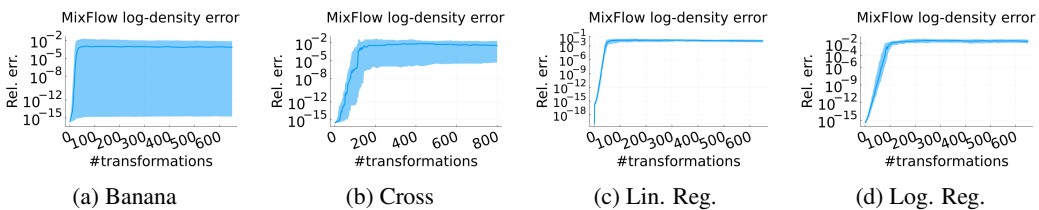

(a) Banana     (b) Cross     (c) Lin. Reg.     (d) Log. Reg.

Figure 5: MixFlow relative log-density evaluation error. The lines indicate the median, and error regions indicate $25^{\mathrm{th}}$ to $75^{\mathrm{th}}$ percentile from 100 evaluations on different positions.

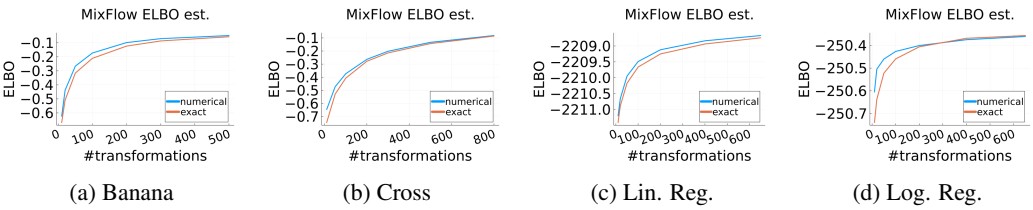

(a) Banana     (b) Cross     (c) Lin. Reg.     (d) Log. Reg.

Figure 6: MixFlow `exact` and `numerical` ELBO estimation over increasing flow length. The lines indicate the averaged ELBO estimates over 200 independent forward orbits. The Monte Carlo error for the given estimates is sufficiently small so we omit the error bar.

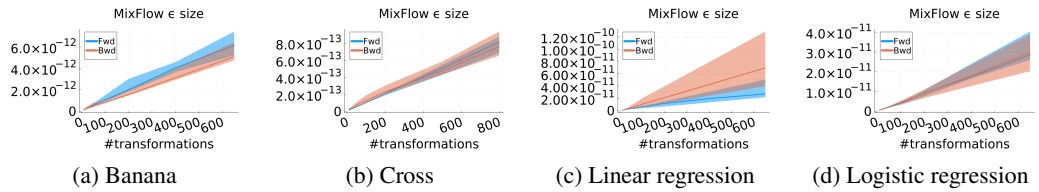

(a) Banana     (b) Cross     (c) Linear regression     (d) Logistic regression

Figure 7: MixFlow shadowing window size $\epsilon$ over increasing flow length. The lines indicate the median, and error regions indicate $25^{\mathrm{th}}$ to $75^{\mathrm{th}}$ percentile from 10 runs.

Finally, since $\epsilon$ depends on $\lambda_{\min}(AA^T)^{-1/2}$, which itself depends on $N$, the shadowing window size may potentially increase with $N$ in practice. Understanding this dependence accurately for a general dynamical system is challenging; there are examples where it remains constant, for instance (see Examples B.1 and B.2 in Appendix B.2), but $\epsilon$ may grow with $N$. Our empirical results in next section show that on representative inferential examples with over 500 flow layers, $\epsilon$ scales roughly linearly with $N$ and is of a similar order of magnitude as the floating point representation error. For further discussion on this topic, we refer readers to Appendix B.2.

## 6 Experiments

In this section, we verify our error bounds and diagnostic procedure of MixFlow on the banana, cross, and 2 real data targets—a Bayesian linear regression and logistic regression posterior; detailed

model and dataset descriptions can be found in Appendix D.1. We also provide a similar empirical investigation for Hamiltonian flow [6] on the same synthetic targets in Appendix C.2.

We begin by assessing the error of trajectory-averaged estimates based on approximate draws. Fig. 4 displays the relative numerical estimation error compared to the exact estimation based on the same initial draw. Although substantial numerical error was observed in orbit computation (see Fig. 16 in Appendix D.3), the relative sample estimate error was around $1\%$ for all four examples, suggesting that the statistical properties of the forward orbits closely resemble those of the exact orbit.

We then focus on the density evaluation. For synthetic examples, we assessed densities at 100 evenly distributed points within the target region (Figs. 2a and 2b). For real data, we evaluated densities at the locations of 100 samples from MCMC method None-U-turn sampler (NUTS); detailed settings for NUTS is described in Appendix D.2. It is evident from the relative error shown in Fig. 5 that the numerical density closely matches the exact density evaluation, with the relative numerical error ranging between $0.1\%$ and $1\%$, which is quite small. The absolute error can be found in Fig. 17 in Appendix D.3, showing that the density evaluation error does not grow as substantially as the orbit computation error (Fig. 16 in Appendix D.3), which aligns with the bound in Theorem 4.6.

Fig. 6 further demonstrates the numerical error for the ELBO estimations (Eq. (4)). In each example, both the averaged exact ELBO estimates and numerical ELBO estimates are plotted against an increasing flow length. Each ELBO curve is averaged over 200 independent forward orbits. Across all four examples, the numerical ELBO curve remain closely aligned with the exact ELBO curve, indicating that the numerical error is small in comparison to the scale of the ELBO value. Moreover, the error does not grow with an increasing flow length, and even presents a decreasing trend as $N$ increases in the two synthetic examples and the Logistic regression example. This aligns well with the error bounds provided in Theorem 4.7.

Finally, Fig. 7 presents the size of the shadowing window $\epsilon$ as the flow length $N$ increases. As noted earlier, $\epsilon$ depends on the flow map approximation error $\delta$ and potentially $N$. We evaluated the size of $\delta$ by calculating the approximation error of a single $F$ or $B$ for 1000 i.i.d. draws from the reference distribution. Boxplots of $\delta$ for all examples can be found in Fig. 18 of Appendix D.3. These results show that in the four examples, $\delta$ is close to the floating point representation error (approximately $10^{-14}$). Thus, we fixed $\delta$ at $10^{-14}$ when computing $\epsilon$. As shown in Fig. 7, $\epsilon$ for both the forward and backward orbits grows roughly linearly with the flow length. This growth is significantly less drastic than the exponential growth of the orbit computation error. And crucially, the shadowing window size is very small—smaller than $10^{-10}$ for MixFlow of length over 1000, which justifies the validity of the downstream statistical procedures. In contrast, the orbit computation errors in the two synthetic examples rapidly reach a magnitude similar to the scale of the target distributions, as shown in Fig. 1.

## 7 Conclusion

This work delves into the numerical stability of variational flows, drawing insights from shadowing theory and introducing a diagnostic tool to assess the shadowing property. Experiments corroborate our theory and demonstrate the effectiveness of the diagnostic procedure. However, the scope of our current analysis is limited to downstream tasks post-training of discrete-time flows. Understanding the impact of numerical instability during the training phase or probing into recent architectures like continuous normalizing flows or neural ODEs [21, 37–40], remains an avenue for further exploration. Additionally, while our theory is centered around error bounds that are proportional to the shadowing window size $\epsilon$, we have recognized that $\epsilon$ can grow with $N$. Further study on its theoretical growth rate is needed. Finally, in our experiments, we employed a basic method to calculate the minimum eigenvalue of $AA^T$. Given the sparsity of this matrix, a deeper exploration into a more efficient computational procedure is merited.

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

# A Proofs

## A.1 Layer-wise error analysis

The layer-wise error analysis follows by recursing the one-step bound on the difference between the exact and numerical flow:

$$
\begin{aligned}
&\|F_N \circ \cdots \circ F_1 x - \hat{F}_N \circ \cdots \circ \hat{F}_1 x\| \\
&\leq \|F_N \circ \cdots \circ F_1 x - F_N \circ \hat{F}_{N-1} \circ \ldots \hat{F}_1 x\| + \|F_N \circ \hat{F}_{N-1} \circ \ldots \hat{F}_1 x - \hat{F}_N \circ \cdots \circ \hat{F}_1 x\| \\
&\leq \mathrm{Lip}(F_N) \|F_{N-1} \circ \cdots \circ F_1 x - \hat{F}_{N-1} \circ \cdots \circ \hat{F}_1 x\| + \delta \\
&\leq \cdots \leq \delta \sum_{n=0}^{N-1} \prod_{j=n+2}^{N} \mathrm{Lip}(F_j).
\end{aligned}
$$

## A.2 Error bounds for normalizing flows

*Proof of Proposition 4.2.* Let $S$ be the random initial state of the orbit that shadows the pseudo-orbit of $X \sim q_0$. By triangle inequality,

$$
\left| \mathbb{E}f(X_N) - \mathbb{E}f(\hat{X}_N) \right| \leq \left| \mathbb{E}f(X_N) - \mathbb{E}f(S_N) \right| + \left| \mathbb{E}f(S_N) - \mathbb{E}f(\hat{X}_N) \right|.
$$

By $\epsilon$-shadowing and $\ell$-Lipshitz continuity of $f$,

$$
\left| \mathbb{E}f(S_N) - \mathbb{E}f(\hat{X}_N) \right| \leq \mathbb{E}\left| f(S_N) - f(\hat{X}_N) \right| \leq \ell \mathbb{E}\|S_N - \hat{X}_N\| \leq \ell\epsilon.
$$

Next

$$
\begin{aligned}
&\sup_{f : |f| \leq U, \mathrm{Lip}(f) \leq \ell} |\mathbb{E}f(X_N) - \mathbb{E}f(S_N)| \\
&\leq \sup_{f : |f| \leq U} |\mathbb{E}f(X_N) - \mathbb{E}f(S_N)| = 2U \mathrm{D}_{\mathrm{TV}}(X_N, S_N) = 2U \mathrm{D}_{\mathrm{TV}}(X, S).
\end{aligned}
$$

The last equality is due to the fact that $X_N$ and $S_N$ are the map of $X$ and $S$, respectively, under the bijection $F_N \circ \cdots \circ F_1$, and the fact that the total variation is invariant under bijections [41, Theorem 1]. $\square$

*Proof of Theorem 4.3.* Let $s \in \mathcal{X}$ be the initial state of the backward shadowing orbit. By triangle inequality,

$$
|\log \hat{q}(x) - \log q(x)| \leq |\log q(x) - \log q(s)| + |\log \hat{q}(x) - \log q(s)|.
$$

For the first term on the right-hand side, note that for some $y$ on the segment from $x$ to $s$,

$$
|\log q(x) - \log q(s)| = \left| \nabla \log q(y)^T (x - s) \right| \leq \epsilon \sup_{\|y - x\| \leq \epsilon} \|\nabla \log q(y)\|. \tag{7}
$$

For the second term,

$$
\log \hat{q}(x) - \log q(s) = |\log q_0(\hat{x}_{-N}) - \log q_0(s_{-N})| + \sum_{n=1}^{N} \left| \log J_{N-(n-1)}(\hat{x}_{-n}) - \log J_{N-(n-1)}(s_{-n}) \right|.
$$

We apply the same technique to bound each term as in Eq. (7). $\square$

*Proof of Theorem 4.4.* By definition, $\mathrm{ELBO}(q\|p) = \mathbb{E}[E(X)]$, and by hypothesis the distribution $q_0$ of the initial flow state is equal to the distribution $\xi_0$ of the shadowing orbit initial state, so $\mathbb{E}[E(X)] = \mathbb{E}[E(S)]$. Hence $\left| \mathrm{ELBO}(q\|p) - \mathbb{E}\left[\hat{E}(X)\right] \right| \leq \mathbb{E}\left[ \left| E(S) - \hat{E}(X) \right| \right]$. Finally,

$$
\begin{aligned}
&\left| E(s) - \hat{E}(x) \right| \\
&= \left| \left( \log p(s_N) - \log q_0(s) + \sum_{n=1}^{N} \log J_n(s_n) \right) - \left( \log p(\hat{x}_N) - \log q_0(x) + \sum_{n=1}^{N} \log J_n(\hat{x}_n) \right) \right| \\
&\leq |\log p(s_N) - \log p(\hat{x}_N)| + |\log q_0(s) - \log q_0(x)| + \sum_{n=1}^{N} |\log J_n(s_n) - \log J_n(\hat{x}_n)|.
\end{aligned}
$$

The proof is completed by bounding each difference with the local Lipschitz constant around the pseudo-orbit times the shadowing window size $\epsilon$, and applying the expectation. $\square$

## A.3 Error bounds for MixFlows

*Proof of Proposition 4.5.* Let $S$ be the initial point of the random shadowing orbit. By the definition of MixFlows,

$$
\left| \mathbb{E}f(Y) - \mathbb{E}f(\hat{Y}) \right| = \left| \mathbb{E} \left[ \frac{1}{N+1} \sum_{n=0}^{N} f(X_n) - f(S_n) + f(S_n) - f(\hat{X}_n) \right] \right|
$$

$$
\leq \left| \mathbb{E} \left[ \frac{1}{N+1} \sum_{n=0}^{N} f(X_n) - f(S_n) \right] \right| + \mathbb{E} \left| \frac{1}{N+1} \sum_{n=0}^{N} f(S_n) - f(\hat{X}_n) \right|
$$

$$
\leq \left| \mathbb{E} \left[ \frac{1}{N+1} \sum_{n=0}^{N} f(X_n) - f(S_n) \right] \right| + \frac{1}{N+1} \sum_{n=0}^{N} \mathbb{E} \left| f(S_n) - f(\hat{X}_n) \right|.
$$

To bound the second term, we apply shadowing and the Lipschitz continuity of $f$ to show that each $\left| f(S_n) - f(\hat{X}_n) \right| \leq \ell\epsilon$, and hence their average has the same bound. For the first term, in the case where $N$ is fixed and finite,

$$
\sup_{f:|f|\leq U, \mathrm{Lip}(f)\leq \ell} \left| \mathbb{E} \left[ \frac{1}{N+1} \sum_{n=0}^{N} f(X_n) - f(S_n) \right] \right| \leq \frac{1}{N+1} \sum_{n=0}^{N} \sup_{f:|f|\leq U, \mathrm{Lip}(f)\leq \ell} \left| \mathbb{E} \left[ f(X_n) - f(S_n) \right] \right|
$$

$$
\leq \frac{1}{N+1} \sum_{n=0}^{N} 2U \frac{1}{2} \sup_{f:|f|\leq 1} \left| \mathbb{E} \left[ f(X_n) - f(S_n) \right] \right|
$$

$$
= \frac{1}{N+1} \sum_{n=0}^{N} 2U \mathrm{D_{TV}} \left( X_n, S_n \right)
$$

$$
= \frac{1}{N+1} \sum_{n=0}^{N} 2U \mathrm{D_{TV}} \left( X, S \right)
$$

$$
= 2U \mathrm{D_{TV}} \left( \xi_0, q_0 \right).
$$

We now consider the large-$N$ limiting case when $F$ is $\pi$-ergodic and measure-preserving, $q_0 \ll \pi$ and $\xi_0 \ll \pi$, and $U = \ell = 1$. Let $Z = S_K$, $K \sim \mathsf{Unif}\{0, 1, \dots, N\}$, and let $W \sim \pi$. In this case, by the triangle inequality,

$$
\sup_{f:|f|\leq U, \mathrm{Lip}(f)\leq \ell} \left| \mathbb{E} \left[ \frac{1}{N+1} \sum_{n=0}^{N} f(X_n) - f(S_n) \right] \right| \leq \mathrm{D_{BL}} \left( Y, W \right) + \mathrm{D_{BL}} \left( Z, W \right).
$$

By [5, Theorem 4.1], both terms on the right-hand side converge to 0 as $N \to \infty$. $\quad\square$

**Lemma A.1.** *If $\forall i \in [n], a_i, b_i > 0$, then*

$$
\left| \log \frac{a_1 + \cdots + a_n}{b_1 + \cdots + b_n} \right| \leq \max_{1 \leq i \leq n} \left\{ \left| \log \frac{a_i}{b_i} \right| \right\}
$$

*Proof.* Define $A := \sum_{i=1}^{n} a_i$,

$$
\log \frac{a_1 + \cdots + a_n}{b_1 + \cdots + b_n} = -\log \frac{b_1 + \cdots + b_n}{a_1 + \cdots + a_n} = -\log \sum_{i=1}^{n} \frac{a_i}{A} \frac{b_i}{a_i}
$$

Jensen's inequality yields that

$$
\log \frac{a_1 + \cdots + a_n}{b_1 + \cdots + b_n} \leq \sum_{i=1}^{n} \frac{a_i}{A} \log \frac{a_i}{b_i} \leq \max_{1 \leq i \leq n} \left\{ \log \frac{a_i}{b_i} \right\}.
$$

Applying the same technique to the other direction yields the result. $\quad\square$

*Proof of Theorem 4.6.* First, shadowing and the triangle inequality yields

$$
| \log \hat{q}(x) - \log q(x) | \leq | \log q(x) - \log q(s) | + | \log \hat{q}(x) - \log q(s) |
$$
$$
\leq \epsilon L_{q,\epsilon}(x) + | \log \hat{q}(x) - \log q(s) |.
$$

By Lemma A.1, we have

$$
| \log \hat{q}(x) - \log q(s) | \leq \max_{0 \leq n \leq N} \left| \log \frac{q_0(\hat{x}_{-n}) / \prod_{j=1}^{n} J(\hat{x}_{-j})}{q_0(s_{-n}) / \prod_{j=1}^{n} J(s_{-j})} \right|
$$

$$
= \max_{0 \leq n \leq N} | \log \hat{q}_n(x) - \log q_n(s) |,
$$

where $q_n$ is the density of the length-$n$ normalizing flow with constant forward flow map $F$, and $\hat{q}_n$ is its approximation via the pseudo-orbit. Using the same technique as in the proof of Theorem 4.3,

$$\max_{0 \leq n \leq N} |\log \hat{q}_n(x) - \log q_n(s)| \leq \max_{0 \leq n \leq N} \epsilon \cdot \left( L_{q_0,\epsilon}(\hat{x}_{-n}) + \sum_{j=1}^{n} L_{J,\epsilon}(\hat{x}_{-j}) \right)$$

$$\leq \epsilon \cdot \left( \max_{0 \leq n \leq N} L_{q_0,\epsilon}(\hat{x}_{-n}) + \sum_{n=1}^{N} L_{J,\epsilon}(\hat{x}_{-n}) \right).$$

Combining this with the earlier bound yields the first stated result. To obtain the second result in the infinite shadowing setting, we repeat the process of bounding $\max_{0 \leq n \leq N} |\log \hat{q}_n(x) - \log q_n(s)|$, but rather than expressing each supremum around the pseudo-orbit $\hat{x}_{-n}$, we express it around the shadowing orbit $s_{-n}$ to find that

$$\max_{0 \leq n \leq N} |\log \hat{q}_n(x) - \log q_n(s)| \leq \epsilon \cdot \left( \max_{0 \leq n \leq N} L_{q_0,\epsilon}(s_{-n}) + \sum_{n=1}^{N} L_{J,\epsilon}(s_{-n}) \right).$$

We then bound $\max_{0 \leq n \leq N}$ with a sum and merge this with the first bound to find that

$$\limsup_{N \to \infty} \frac{1}{N} |\log \hat{q}(x) - \log q(x)| \leq \epsilon \cdot \limsup_{N \to \infty} \left( \frac{1}{N} L_{q,\epsilon}(x) + \frac{1}{N} \sum_{n=0}^{N} L_{q_0,\epsilon}(s_{-n}) + \frac{1}{N} \sum_{n=1}^{N} L_{J,\epsilon}(s_{-n}) \right).$$

Since $L_{q,\epsilon}(x) = o(N)$ as $N \to \infty$, the first term decays to 0. The latter two terms are ergodic averages under the backward dynamics initialized at $s$; if the pointwise ergodic theorem [42], [43, p. 212] holds at $s$, then the result follows. However, the pointwise ergodic theorem holds only $\pi$-almost surely for each of $L_{q_0,\epsilon}$ and $L_{J,\epsilon}$. Denote $\mathcal{Z}$ to be the set of $\pi$-measure zero for which the theorem does not apply. Suppose there is a set $\mathcal{A} \subseteq \mathcal{X}$ such that $q_0(\mathcal{A}) > 0$, and $x \in \mathcal{A}$ implies that $s \in \mathcal{Z}$. But then $\xi_0(\mathcal{Z}) > 0$, which contradicts $\xi_0 \ll \pi$. Therefore the pointwise ergodic theorem applies to $s$ for $q_0$-almost every $x \in \mathcal{X}$. $\qquad \square$

*Proof of Theorem 4.7.* By definition, $\text{ELBO}(q||p) = \mathbb{E}[E(X)]$, and by hypothesis the distribution $q_0$ of the initial flow state is equal to the distribution $\xi_0$ of the joint shadowing orbit initial state, so $\mathbb{E}[E(X)] = \mathbb{E}[E(S)]$. Hence $\left| \text{ELBO}(q||p) - \mathbb{E}[\hat{E}(X)] \right| \leq \mathbb{E}\left[ \left| E(S) - \hat{E}(X) \right| \right]$. So applying the triangle inequality, we find that

$$\left| E(s) - \hat{E}(x) \right| \leq \left| \frac{1}{N+1} \sum_{n=0}^{N} \log \frac{p(s_n)}{p(\hat{x}_n)} \right| + \left| \frac{1}{N+1} \sum_{n=0}^{N} \log \frac{\frac{1}{N+1} \sum_{j=0}^{N} \frac{q_0(\hat{x}_{n-j})}{\prod_{i=1}^{j} J(\hat{x}_{n-i})}}{\frac{1}{N+1} \sum_{j=0}^{N} \frac{q_0(s_{n-j})}{\prod_{i=1}^{j} J(s_{n-i})}} \right|$$

For the first sum, each term is bounded using the local Lipschitz constant,

$$\left| \frac{1}{N+1} \sum_{n=0}^{N} \log \frac{p(s_n)}{p(\hat{x}_n)} \right| \leq \frac{1}{N+1} \sum_{n=0}^{N} |\log p(s_n) - \log p(\hat{x}_n)|$$

$$\leq \frac{1}{N+1} \sum_{n=0}^{N} \epsilon L_{p,\epsilon}(\hat{x}_n).$$

For the second, we apply Lemma A.1 and then use the local Lipschitz constants,

$$\left| \frac{1}{N+1} \sum_{n=0}^{N} \log \frac{\frac{1}{N+1} \sum_{j=0}^{N} \frac{q_0(\hat{x}_{n-j})}{\prod_{i=1}^{j} J(\hat{x}_{n-i})}}{\frac{1}{N+1} \sum_{j=0}^{N} \frac{q_0(s_{n-j})}{\prod_{i=1}^{j} J(s_{n-i})}} \right| \leq \frac{1}{N+1} \sum_{n=0}^{N} \left| \log \frac{\frac{1}{N+1} \sum_{j=0}^{N} \frac{q_0(\hat{x}_{n-j})}{\prod_{i=1}^{j} J(\hat{x}_{n-i})}}{\frac{1}{N+1} \sum_{j=0}^{N} \frac{q_0(s_{n-j})}{\prod_{i=1}^{j} J(s_{n-i})}} \right|$$

$$\leq \frac{1}{N+1} \sum_{n=0}^{N} \max_{0 \leq j \leq N} \left| \log \frac{\frac{q_0(\hat{x}_{n-j})}{\prod_{i=1}^{j} J(\hat{x}_{n-i})}}{\frac{q_0(s_{n-j})}{\prod_{i=1}^{j} J(s_{n-i})}} \right|$$

$$\leq \frac{1}{N+1} \sum_{n=0}^{N} \max_{0 \leq j \leq N} |\log q_0(\hat{x}_{n-j}) - \log q_0(s_{n-j})| + \sum_{i=1}^{j} |\log J(\hat{x}_{n-i}) - \log J(s_{n-i})|$$

$$\leq \epsilon \left( \frac{1}{N+1} \sum_{n=0}^{N} \max_{0 \leq j \leq N} L_{q_0,\epsilon}(\hat{x}_{n-j}) + \sum_{i=1}^{j} L_{J,\epsilon}(\hat{x}_{n-i}) \right)$$

$$\leq \epsilon \left( \max_{-N \leq n \leq N} L_{q_0,\epsilon}(\hat{x}_n) + \frac{1}{N+1} \sum_{n=0}^{N} \sum_{i=1}^{N} L_{J,\epsilon}(\hat{x}_{n-i}) \right)$$

$$\leq \epsilon \left( \max_{-N \leq n \leq N} L_{q_0,\epsilon}(\hat{x}_n) + \sum_{n=-N}^{N} L_{J,\epsilon}(\hat{x}_n) \right).$$

Combining this with the previous bound and taking the expectation yields the first result,

$$\left|\text{ELBO}\,(q||p) - \mathbb{E}\left[\hat{E}(X)\right]\right| \leq \epsilon \cdot \mathbb{E}\left[\frac{1}{N+1}\sum_{n=0}^{N} L_{p,\epsilon}(\hat{X}_n) + \max_{-N \leq n \leq N} L_{q_0,\epsilon}(\hat{X}_n) + \sum_{n=-N}^{N} L_{J,\epsilon}(\hat{X}_n)\right].$$

Once again to arrive at the second result, we redo the analysis but center the supremum Lipschitz constants around the shadowing orbit points, and replace the maximum with a sum, yielding the bound

$$\left|\text{ELBO}\,(q||p) - \mathbb{E}\left[\hat{E}(X)\right]\right| \leq \epsilon \cdot \mathbb{E}\left[\frac{1}{N+1}\sum_{n=0}^{N} L_{p,\epsilon}(S_n) + \max_{-N \leq n \leq N} L_{q_0,\epsilon}(S_n) + \sum_{n=-N}^{N} L_{J,\epsilon}(S_n)\right]$$

$$\leq \epsilon \cdot \mathbb{E}\left[\frac{1}{N+1}\sum_{n=0}^{N} L_{p,\epsilon}(S_n) + \sum_{n=-N}^{N} L_{q_0,\epsilon}(S_n) + \sum_{n=-N}^{N} L_{J,\epsilon}(S_n)\right].$$

Now since $\xi_0 = q_0$ by assumption, $S_n \overset{d}{=} X_n$ for each $n \in \mathbb{N}$, so

$$\left|\text{ELBO}\,(q||p) - \mathbb{E}\left[\hat{E}(X)\right]\right| \leq \epsilon \cdot \mathbb{E}\left[\frac{1}{N+1}\sum_{n=0}^{N} L_{p,\epsilon}(X_n) + \sum_{n=-N}^{N} L_{q_0,\epsilon}(X_n) + \sum_{n=-N}^{N} L_{J,\epsilon}(X_n)\right].$$

We divide by $N$ and take the limit supremum:

$$\limsup_{N\to\infty} \frac{1}{N}\left|\text{ELBO}\,(q||p) - \mathbb{E}\left[\hat{E}(X)\right]\right|$$

$$\leq \epsilon \limsup_{N\to\infty} \cdot \mathbb{E}\left[\frac{1}{N(N+1)}\sum_{n=0}^{N} L_{p,\epsilon}(X_n) + \frac{1}{N}\sum_{n=-N}^{N} L_{q_0,\epsilon}(X_n) + \frac{1}{N}\sum_{n=-N}^{N} L_{J,\epsilon}(X_n)\right].$$

Consider just the term

$$\mathbb{E}\left[\frac{1}{N}\sum_{n=0}^{N} L_{J,\epsilon}(X_n)\right] = \int \frac{1}{N}\sum_{n=0}^{N} L_{J,\epsilon}(F^n x)q_0(\mathrm{d}x) = \int \frac{1}{N}\sum_{n=0}^{N} L_{J,\epsilon}(F^n x)\frac{\mathrm{d}q_0}{\mathrm{d}\pi}(x)\pi(\mathrm{d}x).$$

[43, Theorem 8.10 (vi)] asserts that as long as $L_{J,\epsilon} \in L^{m_1}(\pi)$ and $\frac{\mathrm{d}q_0}{\mathrm{d}\pi} \in L^{m_2}(\pi)$ for $1/m_1 + 1/m_2 = 1$, $1 \leq m_1 < \infty$, then

$$\lim_{N\to\infty} \int \frac{1}{N}\sum_{n=0}^{N} L_{J,\epsilon}(F^n x)\frac{\mathrm{d}q_0}{\mathrm{d}\pi}(x)\pi(\mathrm{d}x) = \int L_{J,\epsilon}(x)\pi(\mathrm{d}x)\int \frac{\mathrm{d}q_0}{\mathrm{d}\pi}\pi(\mathrm{d}x) = \mathbb{E}\left[L_{J,\epsilon}(X)\right], \quad X \sim \pi.$$

Applying this result to each term above yields the stated result. $\qquad\square$

## A.4 Norm of matrix right inverse

Notice that the operator $A$ in Theorem 5.1 has a following matrix representation:

$$A = \begin{bmatrix} -D_1 & I & & \\ & -D_2 & I & \\ & & \ddots & \ddots \\ & & & -D_N & I \end{bmatrix} \in \mathbb{R}^{dN \times d(N+1)}, \text{ where } D_k = \nabla F_k(\hat{x}_{k-1}), \forall k \in [N]. \quad (8)$$

**Proposition A.2** ([33, Corollary 4.2]). *Let $A$ be as defined in Eq. (8). Suppose $(F_n)_{n=1}^{N}$ are all differentiable. Then*

$$A^{\dagger} := A^T(AA^T)^{-1} \in \arg\min_{X}\|X\|, \quad \text{subject to } AX = I, \quad (9)$$

*and $\|A^{\dagger}\| = \sigma_{\min}(A)^{-1}$, where $\sigma_{\min}(A)$ denotes the smallest singular value of $A$.*

*Proof of Proposition A.2.* Note that $A$ has full row rank due to the identity blocks. In this case, $AA^T$ is invertible, thus $A^T(AA^T)^{-1}$ is a valid right inverse of $A$. Eq. (9) is then obtained by a direct application of [33, Corollary 4.2].

To obtain the norm of $A^{\dagger}$, since $A$ has full row rank,

$$A = U\Sigma V^T, \quad \text{where } U \in \mathbb{R}^{dN \times dN}, V \in \mathbb{R}^{d(N+1) \times d(N+1)} \text{ are orthonormal matrices,}$$

$$\Sigma \in \mathbb{R}^{dN \times d(N+1)} \text{ is a rectangular diagonal matrix with full row rank.}$$

Then

$$\|A^\dagger\| = \left\| V\Sigma^T U^T \left( U\Sigma V^T V\Sigma^T U^T \right)^{-1} \right\|$$

$$= \left\| V\Sigma^T U^T U(\Sigma\Sigma^T)^{-1} U^T \right\|$$

$$= \|\Sigma(\Sigma\Sigma^T)^{-1}\|$$

$$= \sigma_{\min}(A)^{-1}$$

Notice that since $AA^T$ is an invertible Hermitian matrix, $\sigma_{\min}(A)$ is strictly positive. □

## B  Discussion about the shadowing window $\epsilon$

### B.1  Numerical error when computing $\epsilon$

As mentioned in Section 5, due to the floating-point error $\delta$ involved in evaluating $\nabla F_k$, the evaluation of $A$ is perturbed by the numerical error. This consequently leads to an inaccuracy of the estimation of $\lambda = \|A_r^{-1}\|$. Here $A_r^{-1}$ denotes the right inverse of $A$. In this section, we explain how to calibrate the calculation of $\lambda$ to take into account of this error.

This computational challenge, rooted in numerical inaccuracies, was previously acknowledged in Coomes et al. [17, Section 3.4], with a provided remedy. Suppose we compute terms in $\nabla F_k(\hat{x}_{k-1})$ with a floating-point error $\delta$. This introduces $O(\sqrt{N}\delta)$ Frobenius norm error into the practically calculated $A$. Here we denote the digital computed $A$ by $\tilde{A}$, $\tilde{\lambda} := \|\tilde{A}_r^{-1}\|$, and $\tilde{\delta} := \|A - \tilde{A}\|$. According to Coomes et al. [17, Eq. 45], we can employ the following upper bound on $\lambda$:

$$\lambda \leq (1 - \tilde{\delta}\tilde{\lambda})^{-1}\tilde{\lambda} = \left( 1 - O(\sqrt{N\delta})\tilde{\lambda} \right)^{-1} \tilde{\lambda}.$$

Substituting values of $\delta$, $N$, and computed $\tilde{\lambda}$ from our experiments (refer to Fig. 7), this amounts to an inconsequential relative error of around $(1 - 10^{-9})^{-1} \approx 1$ on $\lambda$.

It is also worth discussing the the shadowing window computation methods proposed in Coomes et al. [17, Section 3.4]. As opposed to our generic eigenvalue computation introduced in Section 5, the computation procedure presented in Coomes et al. [17, Section 3.4] can potentially offer better scalability. However, the methodology in Coomes et al. [17] operates under the heuristic that a dynamical system with shadowing is intrinsically hyperbolic, or nearly so. Specifically, the "hyperbolic splitting" discussed in Coomes et al. [44, P.413, Appendix B] (i.e., the choice of $\ell$), and the "hyperbolic threshold" choice in Coomes et al. [44, P.415, Appendix B] (i.e., the choice of $p$), both implicitly assume that the dynamics either exhibit hyperbolicity or tend closely to it. Such presuppositions might be overly restrictive for generic variational flows, given that the underlying dynamics are general time-inhomogeneous systems. When the underlying dynamics are not (nearly) hyperbolic, applying strategies from Coomes et al. [17] could potentially lead to a poor estimation of the shadowing window.

### B.2  Dependence of $\epsilon$ on $N$

One may notice that $\epsilon$ depends on $N$ implicitly through $\lambda_{\min}(AA^T)^{-1/2}$. Recall that $A$ can be represented as a matrix

$$A = \begin{bmatrix} -D_1 & I & & & \\ & -D_2 & I & & \\ & & \ddots & \ddots & \\ & & & -D_N & I \end{bmatrix} \in \mathbb{R}^{dN \times d(N+1)}, \text{ where } D_k = \nabla F_k(\hat{x}_{k-1}), \forall k \in [N].$$

And the shadowing window is given by $\epsilon = 2\delta\lambda_{\min}(AA^T)^{-1/2}$. If $\lambda_{\min}(AA^T)^{-1/2}$ scales badly with $N$, our $O(N\epsilon)$ error bounds become vacuous. Therefore, in this section, we collect some insights about the scaling of $\epsilon$ over increasing $N$. It is important to note that understanding the explicit quantitative relationship between $\epsilon$ and $N$ for general nonautonomous dynamical systems is quite challenging. Although current literature of dynamical systems focuses on studying conditions so that $\epsilon$ does not scale with $N$, these conditions are typically difficult to verify in practice or are only applicable in restrictive settings such as time-homogeneous linear dynamical systems. For a more systematic study, see [17, 20, 28]. The examples provided below merely serve as synthetic illustrations for intuition and should not be taken as definitive statements for all cases.

**Example B.1** (Scaling map in $\mathbb{R}$). *Consider the time-homogeneous dynamics formed by $T : \mathbb{R} \mapsto \mathbb{R}$ such that $\forall x \in \mathbb{R}, T(x) = Cx$ for some $C > 0$. Then the corresponding shadowing window of a $\delta$-pseudo-orbit is*

$$\epsilon = 2\delta \left[ (C-1)^2 + 2C \left( 1 - \cos\frac{\pi}{N+1} \right) \right]^{-\frac{1}{2}}.$$

*Therefore, for all $C \neq 1$, as $N \to \infty$, $\epsilon = O(\delta)$. For $C = 1$, $\epsilon = O(N\delta)$. The formula for $\lambda_{\min}(AA^T)$ above arises because in this case, $AA^T$ has the following symmetric tridiagonal form,*

$$AA^T = \begin{bmatrix} C^2 + 1 & -C & & & \\ -C & C^2 + 1 & -C & & \\ & \ddots & \ddots & \ddots & \\ & & -C & C^2 + 1 & -C \\ & & & -C & C^2 + 1 \end{bmatrix},$$

*whose eigenvalues have the following closed form [45, Page. 19–20]:*

$$\lambda_k(AA^T) = C^2 + 1 - 2C \cos \frac{k\pi}{N+1}, \quad k = 1, \ldots, N.$$

$\triangleleft$

**Example B.2** (Uniform hyperbolic linear map[46, Theorem 12.3 and p. 171])**.** *Consider the time-homogeneous linear dynamical system in $\mathbb{R}^d$, $T : x \mapsto Ax$, where $A \in \mathbb{R}^{d \times d}$ is similar to a block-diagonal matrix $\mathrm{diag}(A_1, A_2)$ such that $\|A_1\| \leq \lambda$ and $\|A_2^{-1}\| \leq \lambda$ for some $\lambda \in (0, 1)$. Then for all $N \in \mathbb{N}$, any $\delta$-pseudo-orbit is $\epsilon$-shadowed, with $\epsilon = \frac{1+\lambda}{1-\lambda}\delta$.* $\triangleleft$

As illustrated in Examples B.1 and B.2, it is possible for the $\epsilon$-shadowing property to hold in a way that $\epsilon$ does not depend on the flow length $N$ and remains in a similar magnitude as $\delta$. This phenomenon is referred to as the *infinite shadowing property* [17, 19, 32] as mentioned in the beginning of Section 4.3. The infinite shadowing property is shown to be *generic* in the space of all dynamical systems [32], meaning that dynamical systems exhibiting this property form a dense subset of the collection of all dynamical systems. However, while genericity offers valuable insights into the typical behavior of dynamical systems, it does not imply that all systems possess the property. As previously explained, only a small class of systems has been shown to have the infinite shadowing property, and verifying this property for a specific given system remains a challenging task.

## C Normalizing flow via Hamiltonian dynamics

While past work on normalizing flows [2, 3, 24–26] typically builds a flexible flow family by composing numerous parametric flow maps that are agnostic to the target distribution, recent target-informed variational flows focus on designing architectures that take into account the structures of the target distribution. A common approach for constructing such target-informed flows is to use Langevin and Hamiltonian dynamics as part of the flow transformations [6, 7, 47–50]. In this section, we begin with a concise overview of two Hamiltonian-based normalizing flows: Hamiltonian variational auto-encoder (HVAE) [7] and Hamiltonian flow (HamFlow) [6]. Subsequently, we present corollaries derived from Theorems 4.3 and 4.4, specifically applying these general results to HVAE and HamFlow. Finally, we offer empirical evaluations of our theory for HamFlow on the same synthetic examples as those examined for MixFlow.

Hamiltonian dynamics (Eq. (10)) describes the evolution of a particle's position $\theta_t$ and momentum $\rho_t$ within a physical system. This movement is driven by a differentiable negative potential energy $\log p(\theta_t)$ and kinetic energy $\frac{1}{2}\rho_t^T \rho_t$:

$$\frac{\mathrm{d}\rho_t}{\mathrm{d}t} = \nabla \log \pi (\theta_t) \quad \frac{\mathrm{d}\theta_t}{\mathrm{d}t} = \rho_t. \tag{10}$$

For $t \in \mathbb{R}$, we define the mappings $H_t : \mathbb{R}^{2d} \to \mathbb{R}^{2d}$, transforming $(\theta_s, \rho_s) \mapsto (\theta_{t+s}, \rho_{t+s})$ according to the dynamics of Eq. (10). Intuitively, traversing the trajectory of $H_t, t \in \mathbb{R}$ allows particles to efficiently explore the distribution of interest's support, thereby motivating the use of $H_t$ as a flow map. In practice, the exact Hamiltonian flow $H_t$ is replaced by its numerical counterpart—leapfrog integration—which involves $K$ iterations of the subsequent steps:

$$\hat{\rho} \leftarrow \rho + \frac{\epsilon}{2}\nabla \log \pi (\theta) \qquad \theta \leftarrow \theta + \epsilon\hat{\rho} \qquad \rho \leftarrow \hat{\rho} + \frac{\epsilon}{2}\nabla \log \pi (\theta).$$

Here $\epsilon$ denotes the step size of the leapfrog integrator. We use $T_{K,\epsilon} : \mathbb{R}^{2d} \to \mathbb{R}^{2d}$ to define the map by sequencing the above three steps $K$ times. There are two key properties of $T_{K,\epsilon}$ that make it suitable for normalizing flow constructions. $T_{K,\epsilon}$ is invertible (i.e., $T_{K,\epsilon}^{-1} = T_{K,-\epsilon}$) and has a simple Jacobian determinant of the form: $|\det \nabla T_{K,\epsilon}| = 1$. This enables a straightforward density transformation formula: for any density $q_0(\theta, \rho)$ on $\mathbb{R}^{2d}$, pushforward of $q$ under $T_{K,\epsilon}$ has the density $q_0(T_{K,-\epsilon}(\theta, \rho))$.

Given the above properties, Caterini et al. [7] and Chen et al. [6] developed flexible normalizing flows—HVAE and HamFlow, respectively—utilizing flow transformations through alternating compositions of multiple $T_{K,\epsilon}$ and linear transformations. Notably, the momentum variable $\rho$ introduced in the Hamiltonian dynamics Eq. (10) necessitates that both HVAE and HamFlow focus on approximating the *augmented target distribution*

$\bar{\pi}(\theta, \rho) \propto \pi(\theta) \cdot \exp(-\frac{1}{2}\rho^T \rho)$ on $\mathbb{R}^{2d}$. This represents the joint distribution of the target $\pi$ and an independent momentum distribution $\mathcal{N}(0, I_d)$.

Specifically, flow layers of HVAE alternate between the leapfrog transformation $T_{K,\epsilon}$ and a momentum scaling layer:

$$(\theta, \rho) \mapsto (\theta, \gamma\rho), \quad \gamma \in \mathbb{R}^+.$$

And those of HamFlow alternate between $T_{K,\epsilon}$ and a momentum standardization layer:

$$(\theta, \rho) \mapsto (\theta, \Lambda(\rho - \mu)), \quad \Lambda \in \mathbb{R}^{d \times d} \text{ is lower triangular with positive diagonals }, \mu \in \mathbb{R}^d.$$

## C.1 Error analysis for Hamiltonian-based flows

We have provided error analysis for generic normalizing flows in Section 4.2. In this section, we specialize the error bounds for density evaluation (Theorem 4.3) and ELBO estimation (Theorem 4.4) for the two Hamiltonian-based flows. $\pi$ denotes the target distribution of interest on $\mathbb{R}^d$, and $\bar{p}$ denotes the augmented target distribution on $\mathbb{R}^{2d}$. $p$ and $\bar{p}$ respectively denote the unnormalized density of $\pi$ and $\bar{\pi}$.

**Corollary C.1.** *Let $q$ be either* HVAE *or* HamFlow *of $N$ layers with $q_0 = \mathcal{N}(0, I_{2d})$. Suppose the backward dynamics has the $(\epsilon, \delta)$-shadowing property. Then*

$$\forall x \in \mathbb{R}^{2d}, \quad |\log \hat{q}(x) - \log q(x)| \leq \epsilon \cdot (L_{q,\epsilon}(x) + \|\hat{x}_{-N}\|) + \epsilon^2.$$

*Proof of Corollary C.1.* Since the Jacobian determinant of each HVAE/HamFlow layer is constant,

$$L_{J_n,\epsilon}(x) = 0, \qquad \forall n \in [N] \text{ and } x \in \mathbb{R}^{2d}. \tag{11}$$

And since $q_0$ is a standard normal distribution,

$$L_{q_0,\epsilon} = \sup_{\|y-x\| \leq \epsilon} \|y\| \leq \|x\| + \epsilon. \tag{12}$$

The proof is then complete by directly applying Eqs. (11) and (12) to the error bound stated in Theorem 4.3. $\qquad \square$

**Corollary C.2.** *Let $q$ be either* HVAE *or* HamFlow *of $N$ layers with $q_0 = \mathcal{N}(0, I_{2d})$. Suppose the forward dynamics has the $(\epsilon, \delta)$-shadowing property, and $\xi_0 = q_0$. Then*

$$\left| \text{ELBO}(q\|\bar{p}) - \mathbb{E}[\hat{E}(X)] \right| \leq \epsilon \cdot \left( \mathbb{E}\left[ L_{\bar{p},\epsilon}(\hat{X}_N) \right] + \sqrt{2d} \right) + \epsilon^2 \qquad \text{for } X \sim q_0.$$

*If additionally $\log p$ is $L$-smooth (i.e., $\forall x, y, \|\nabla \log p(x) - \nabla \log p(y)\| \leq L\|x - y\|$), then*

$$\left| \text{ELBO}(q\|\bar{p}) - \mathbb{E}[\hat{E}(X)] \right| \leq \epsilon \cdot \left( \mathbb{E}\left[ \|\nabla \log \bar{p}(\hat{X}_N)\| \right] + \sqrt{2d} \right) + (L \vee 1 + 1)\epsilon^2 \qquad \text{for } X \sim q_0.$$

*Proof of Corollary C.2.* Similar to the proof of Fig. 2d, applying Eqs. (11) and (12) to the error bound stated in Theorem 4.4 yields that

$$\left| \text{ELBO}(q\|\bar{p}) - \mathbb{E}[\hat{E}(X)] \right| \leq \epsilon \cdot \mathbb{E}\left[ L_{\bar{p},\epsilon}(\hat{X}_N) + \|X\| \right] + \epsilon^2 \qquad \text{for } X \sim q_0.$$

The first result is then obtained by employing the following bound via Jensen's inequality:

$$\mathbb{E}[\|X\|] \leq \sqrt{\mathbb{E}[\|X\|^2]} = \sqrt{2d} \qquad \text{for } X \sim q_0 = \mathcal{N}(0, I_{2d}).$$

To arrive at the second result, we apply the following bound on $L_{p,\epsilon}$ using the fact that $\nabla \log p$ is $L$-Lipschitz continuous:

$$L_{\bar{p},\epsilon} \leq \|\nabla \log \bar{p}(x)\| + \sup_{\|y-x\| \leq \epsilon} \|\nabla \log \bar{p}(x) - \nabla \log \bar{p}(y)\| \leq \|\nabla \log \bar{p}(x)\| + L \vee 1 \cdot \epsilon.$$

$\qquad \square$

## C.2 Empirical results for Hamiltonian flow

In this section, we empirically validate our error bounds and the diagnostic procedure of HamFlow using two synthetic targets: the banana and cross distributions. We employ a warm-started HamFlow, as detailed in [6, Section 3.4]. This is achieved through a single pass of 10,000 samples from the reference distribution, $q_0$, without additional optimization. Specifically, the $\mu$ and $\Lambda$ values for each flow standardization layer are determined by the sample mean and the Cholesky factorization of the sample covariance, respectively, of the batched momentum samples processed through the layer. For the banana distribution, we set the leapfrog step $K$ at 200 and a step size $\epsilon$ of 0.01. For the cross distribution, we set $K$ to 100 with an $\epsilon$ value of 0.02. Figs. 8a

and 8b showcase a comparison between 1000 i.i.d. samples from the warm-started HamFlow with 400 layers and the true targets. The results illustrate that HamFlow generates samples of decent quality.

We start by examining the numerical orbit computaiton error. As demonstrated in Fig. 9, the pointwise evaluation error grows exponentially with increasing flow depth, and quickly reaches the same order of magnitude as the scale of the target distributions. However, similar to what we have observed in the experiments of MixFlow (described in Section 6), the substantial numerical error of orbit computation does not impact the accuracy of Monte Carlo estimations of expectations and ELBO estimations significantly. Fig. 10 shows the relative numerical error of Monte Carlo estimates of the expectations of three test functions. The relative samples estimation error remains within 5% for the two bounded Lipschitz continuous test functions, aligning with our error bound in Proposition 4.2. Moreover, even for the unbounded test function, the error stays within reasonable limits: 10% for the banana and 5% for the cross examples. Fig. 11 then compares the numerical ELBO estimates and exact ELBO estimates for HamFlow over increasing flow length. In both examples, the numerical ELBO curve is almost identical to the exact one, notwithstanding the considerable sample computation error of the forward orbit.

However, unlike the small log-density evaluation error we observed in the MixFlow experiments, HamFlow's log-density shows significant issues because of the numerical errors of backward orbits computation. In Fig. 12, we measure the numerical error for the log-density evaluation of HamFlow at 100 i.i.d. points from the target distribution. As we add more flow layers, the error grows quickly. For the banana example, the log-density evaluation error can even go above 1000. This raises two main questions: (1) Why is the ELBO estimation error different from the log-density evaluation error in HamFlow? (2) Why is the scaling of the log-density evaluation error different between MixFlow and HamFlow? Our theory presented in Section 4 and Appendix C.1 offers explanation to these questions.

To address the first question, we compare the bounds for HamFlow on log-density evaluation (Corollary C.1) and ELBO estimation (Corollary C.2). The main difference between these bounds is in the local Lipschitz constants. The error in log-density evaluation is based on $L_{q,\epsilon}$ while the error in ELBO estimation is based on $\mathbb{E}[L_{\bar{p},\epsilon}]$. These two quantities— $L_{q,\epsilon}$ and $\mathbb{E}[L_{\bar{p},\epsilon}]$—relate to the smoothness of the learned log-density of HamFlow and log-density of the target. Figs. 8c to 8f display $\log p$ and $\log q$ side-by-side for both synthetic examples. Indeed, in both cases, $\log q$ markedly deviates from $\log p$ and presents drastic fluctuations, indicating that $L_{q,\epsilon}$ can be substantially larger than $\mathbb{E}[L_{\bar{p},\epsilon}]$. Fig. 13 further provides a quantitative comparison on how $L_{q,\epsilon}$ and $\mathbb{E}[L_{\bar{p},\epsilon}]$ change as we increase the flow length. Note that since both $L_{q,\epsilon}$ and $L_{\bar{p},\epsilon}$ are intractable in general, we instead focus on the scaling of $\|\nabla \log q\|$ (which is a lower bound of $L_{q,\epsilon}$) and an upper bound of $\mathbb{E}[L_{\bar{p},\epsilon}]$ (derived from case-by-case analysis for both the banana and cross distributions; see Appendix C.3 for detailed derivation). Results show that in both synthetic examples, $L_{q,\epsilon}$ can increase exponentially over increasing flow length and is eventually larger than $\mathbb{E}[L_{\bar{p},\epsilon}]$ by 30 orders of magnitude, while $\mathbb{E}[L_{\bar{p},\epsilon}]$ does not grow and remains within a reasonable scale.

Similarly, the differences in log-density errors between MixFlow and HamFlow come from how smooth the learned log-density is in each method, according to the error bounds presented in Theorems 4.3 and 4.6. Fig. 2 visualizes the learned log-densities of MixFlow, which matches closely to the actual target distribution, and has a smoother output than HamFlow. This matches what was noted in Xu et al. [5, Figure 3], where MixFlow was shown to approximate the target density better than standard normalizing flows.

Finally, Fig. 14 presents the single step evaluation error $\delta$ of HamFlow and the size of the shadowing window $\epsilon$ as the flow length $N$ increases. Results show that for both synthetic examples, $\delta$ is approximately $10^{-7}$, hence we fix $\delta$ to be this value when computing the shadowing window $\epsilon$. Compared to the significant orbit evaluation error when $N$ is large, the size of $\epsilon$ remains small for both the forward and backward orbits (in the scale of $10^{-4}$ to $10^{-3}$ when $N = 400$). And more importantly, the size of $\epsilon$ grows less drastically than the orbit computation error with the flow length—roughly quadratically in the banana example, and linearly in the cross example, which is similar to what is observed in the MixFlow experiments.

## C.3 Upper bounds of $L_{\bar{p},\epsilon}$ for synthetic examples

Recall that we denote $\bar{p}(x) \propto p([x]_{1:2}) \cdot \exp(-\frac{1}{2}[x]_{3:4}^T [x]_{3:4})$ the augmented target and $p$ denotes the 2d-synthetic target we are interested.

Notice that

$$L_{\bar{p},\epsilon}(x) = \sup_{\|y-x\| \leq \epsilon} \|\nabla \log p([y]_{1:2}) - [y]_{3:4}\| \leq L_{p,\epsilon}([x]_{1:2}) + \|[x]_{3:4}\| + \epsilon,$$

where $L_{p,\epsilon}$ requires a case-by-case analysis. Therefore, in this section, we focus on bounding the local Lipschitz constant for $\log p$.

**Proposition C.3.** *For the 2-dimensional banana distribution, i.e.,*

$$p(x) \propto \exp\left(-\frac{1}{2}\frac{x_1^2}{\sigma^2} - \frac{1}{2}\left(x_2 + bx_1^2 - \sigma^2 b\right)\right),$$

*we have that for all $x =: \begin{bmatrix} x_1 \\ x_2 \end{bmatrix} \in \mathbb{R}^2$,*

$$L_{p,\epsilon}(x) \leq \|\nabla \log p(x)\| + \epsilon \cdot \left( 2b|x_1| + \max\left\{ \left| 6b^2 x_1^2 + 2bx_2 - 2\sigma^2 b^2 + \frac{1}{\sigma^2} \right|, 1 \right\} + 2b \max\left\{ 6b|x_1| + 6b(\epsilon + \epsilon^2), 1\epsilon \right\} \right)$$

*Proof of Proposition C.3.* By the mean value theorem,

$$L_{p,\epsilon}(x) \leq \|\nabla \log p(x)\| + \sup_{\|y-x\| \leq \epsilon} \|\nabla \log p(y) - \nabla \log p(x)\|$$

$$\leq \|\nabla \log p(x)\| + \sup_{\|y-x\| \leq \epsilon} \|\nabla^2 \log p(y)\| \cdot \epsilon.$$

The proof will then complete by showing that

$$\sup_{\|y-x\| \leq \epsilon} \|\nabla^2 \log p(y)\| \leq 2b|x_1| + \max\left\{ \left| 6b^2 x_1^2 + 2bx_2 - 2\sigma^2 b^2 + \frac{1}{\sigma^2} \right|, 1 \right\} + 2b \max\left\{ 6b|x_1| + 6b(\epsilon + \epsilon^2), \epsilon \right\}.$$

$$(13)$$

To verify Eq. (13), note that

$$\nabla^2 \log p(x) = -\begin{bmatrix} 6b^2 x_1^2 + 2bx_2 - 2\sigma^2 b^2 + \frac{1}{\sigma^2} & 2bx_1 \\ 2bx_1 & 1 \end{bmatrix}$$

The Gershgorin circle theorem then yields that

$$\|\nabla^2 \log p(x)\| \leq 2b|x_1| + \max\left\{ \left| 6b^2 x_1^2 + 2bx_2 - 2\sigma^2 b^2 + \frac{1}{\sigma^2} \right|, 1 \right\}.$$

Additionally, we have

$$\sup_{\|y_1 - x_1\| \leq \epsilon} |y_1| \leq |x_1| + \epsilon$$

and

$$\sup_{\|y-x\| \leq \epsilon} \left| 6b^2 y_1^2 + 2by_2 - 2\sigma^2 b^2 + \frac{1}{\sigma^2} \right| \leq \left| 6b^2 x_1^2 + 2bx_2 - 2\sigma^2 b^2 + \frac{1}{\sigma^2} \right| + 2b \max\left\{ 6b|x_1| + 6b(\epsilon + \epsilon^2), \epsilon \right\}.$$

The proof is then completed. $\qquad \square$

**Proposition C.4.** *For the cross distribution, a 4-component Gaussian mixture of the form*

$$p(x) = \frac{1}{4} \mathcal{N}\left( x \Big| \begin{bmatrix} 0 \\ 2 \end{bmatrix}, \begin{bmatrix} 0.15^2 & 0 \\ 0 & 1 \end{bmatrix} \right) + \frac{1}{4} \mathcal{N}\left( x \Big| \begin{bmatrix} -2 \\ 0 \end{bmatrix}, \begin{bmatrix} 1 & 0 \\ 0 & 0.15^2 \end{bmatrix} \right)$$

$$+ \frac{1}{4} \mathcal{N}\left( x \Big| \begin{bmatrix} 2 \\ 0 \end{bmatrix}, \begin{bmatrix} 1 & 0 \\ 0 & 0.15^2 \end{bmatrix} \right) + \frac{1}{4} \mathcal{N}\left( x \Big| \begin{bmatrix} 0 \\ -2 \end{bmatrix}, \begin{bmatrix} 0.15^2 & 0 \\ 0 & 1 \end{bmatrix} \right),$$

*we have that for all $x \in \mathbb{R}^2$,*

$$L_{p,\epsilon}(x) \leq \frac{1}{0.15^2} \left( 2 + \|x\| + \epsilon \right).$$

*Proof of Proposition C.4.* For notational convenience, denote the density of 4 Gaussian component by $p_1, p_2, p_3, p_4$, hence

$$\log p(x) = \log \{ p_1(x) + p_2(x) + p_3(x) + p_4(x) \} - \log 4.$$

Examining $\nabla \log p(x)$ directly yields that for all $x \in \mathbb{R}^2$,

$$\nabla \log p(x) = \frac{1}{p_1(x) + p_2(x) + p_3(x) + p_4(x)} \sum_{i=1}^{4} p_i(x) \Sigma_i^{-1}(\mu_i - x)$$

$$\in \text{conv}\left\{ \Sigma_i^{-1}(\mu_i - x) : i = 1, 2, 3, 4 \right\}.$$

Here $\mu_i$ and $\Sigma_i$ denotes the Gaussian mean and covariance of $p_i$. Then by Jensen's inequality,

$$\sup_{\|y-x\| \leq \epsilon} \|\nabla \log p(y)\| \leq \max\left\{ \sup_{\|y-x\| \leq \epsilon} \|\Sigma_i^{-1}(\mu_i - y)\| : i = 1, 2, 3, 4 \right\}$$

$$\leq \max_{i \in [4]} \left\{ \|\Sigma_i^{-1}\| \cdot \left( \|\mu_i\| + \sup_{\|y-x\| \leq \epsilon} \|y\| \right) \right\}$$

$$\leq \frac{1}{0.15^2} \left( 2 + \|x\| + \epsilon \right).$$

$$\qquad \square$$

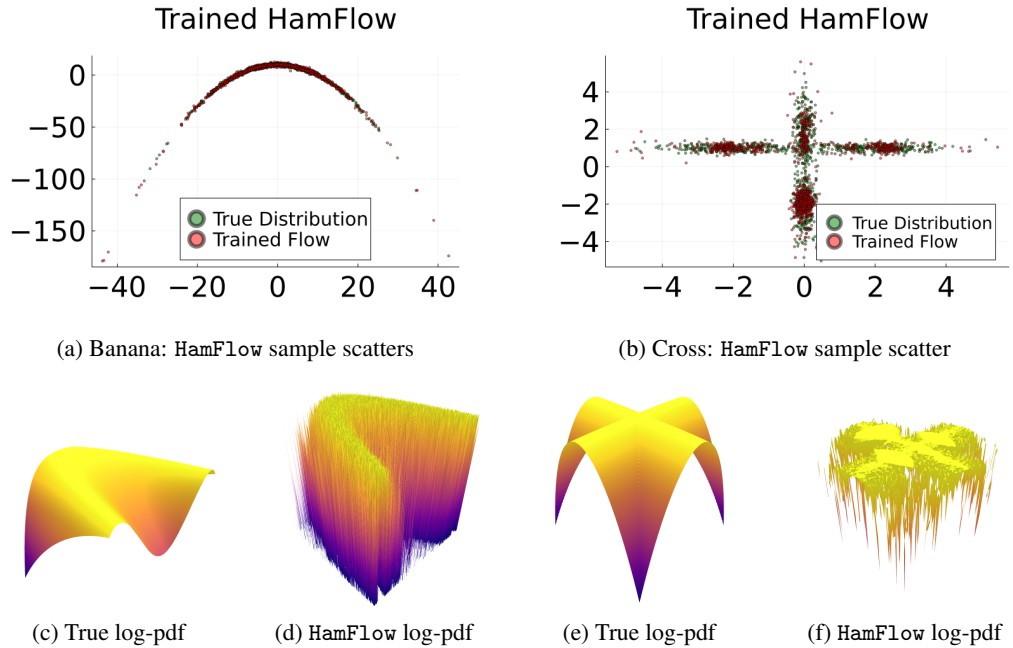

(a) Banana: `HamFlow` sample scatters  (b) Cross: `HamFlow` sample scatter

(c) True log-pdf  (d) `HamFlow` log-pdf  (e) True log-pdf  (f) `HamFlow` log-pdf

Figure 8: Visualization of sample scatters and density of `HamFlow`. Figure (a)–(b) each shows 1000 i.i.d. draws from `HamFlow` distribution (red) and target distribution (green). (c) and (e) show respectively the exact log-density of the Banana and the cross distribution, while (d) and (f) show respectively the sliced `HamFlow` log-density evaluation at $\rho = (0, 0)$ on those two targets. The `HamFlow` log-density evaluations are computed via $2048$-big `BigFloat` representation.

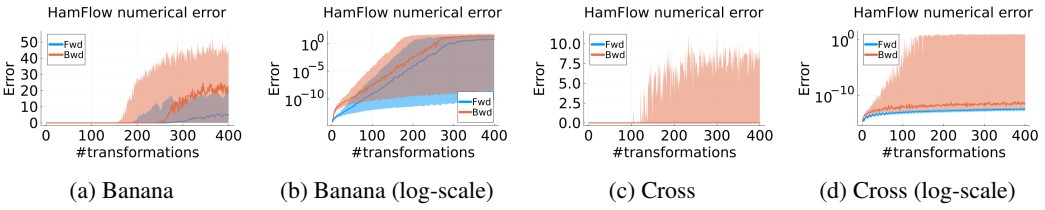

(a) Banana  (b) Banana (log-scale)  (c) Cross  (d) Cross (log-scale)

Figure 9: `HamFlow` forward (`fwd`) and backward (`bwd`) orbit errors on synthetic examples. (a) and (c) show the median and upper/lower quartile forward error $\|F^k x - \hat{F}^k x\|$ and backward error $\|B^k y - \hat{B}^k y\|$ comparing $k$ transformations of the forward exact/approximate maps $F$, $\hat{F}$ or backward exact/approximate maps $B$, $\hat{B}$. And (b) and (d) respectively display (a) and (c) in a logarithmic scale, intended to provide a clearer illustration of the exponential growth of the error. The lines indicate the median, and error regions indicate $25^{\text{th}}$ to $75^{\text{th}}$ percentile from 100 independent runs, which take an i.i.d. draw of $x$ from $q_0$ and draw $y$ from the actual target distribution $p$.

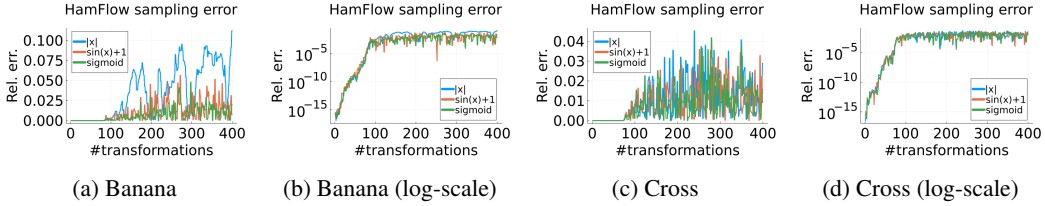

(a) Banana  (b) Banana (log-scale)  (c) Cross  (d) Cross (log-scale)

Figure 10: `HamFlow` relative sample average computaion error on three test functions: $\sum_{i=1}^{d} [|x|]_i, \sum_{i=1}^{d} [\sin(x) + 1]_i$ and $\sum_{i=1}^{d} [\text{sigmoid}(x)]_i$. Each line denotes the numerical error computed from Monte Carlo average of the three test functions via 100 independent forward orbits starting at i.i.d. draws from $q_0$.

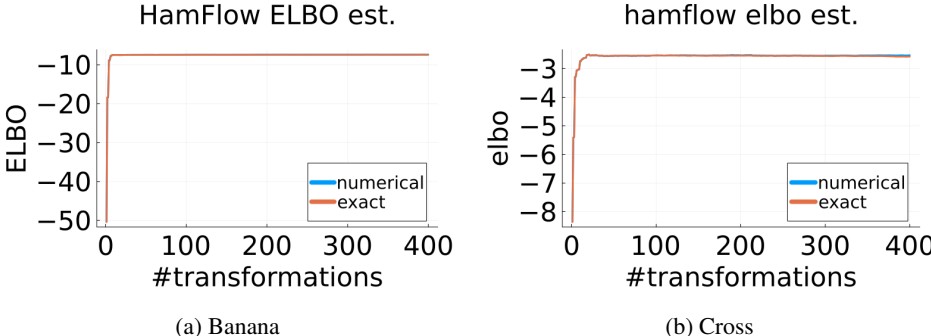

(a) Banana

(b) Cross

Figure 11: HamFlow `exact` and `numerical` ELBO estimation over increasing flow length. The lines indicate the averaged ELBO estimates over 100 independent forward orbits. The Monte Carlo error for the given estimates is sufficiently small so we omit the error bar.

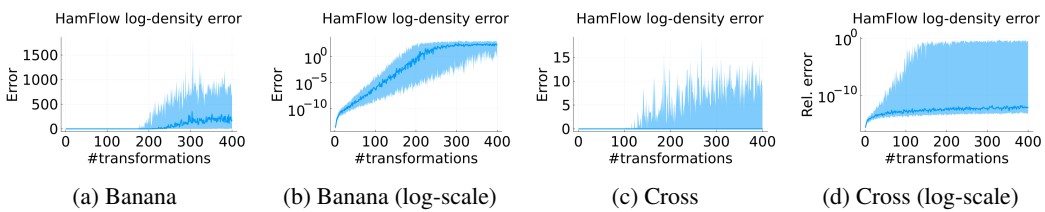

(a) Banana  (b) Banana (log-scale)  (c) Cross  (d) Cross (log-scale)

Figure 12: HamFlow log-density evaluation error on synthetic examples. Log-densities are assessed at positions of 100 independent samples from the target distribution. (b) and (d) respectively display (a) and (c) in a logarithmic scale for a better illustration of the growth of error over increasing flow length. The lines indicate the median, and error regions indicate $25^{\text{th}}$ to $75^{\text{th}}$ percentile from the 100 evaluations.

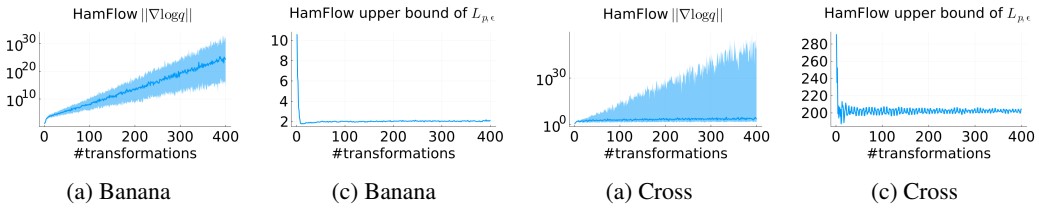

(a) Banana  (c) Banana  (a) Cross  (c) Cross

Figure 13: Comparison of HamFlow's $L_{q,\epsilon}$ and $\mathbb{E}[L_{\bar{p},\epsilon}(\hat{X}_N)]$ on the banana and cross distributions. Figs. 13a and 13c plot $\|\nabla \log q(x)\|$ against flow length $N$; the lines indicate the median, and error regions indicate $25^{\text{th}}$ to $75^{\text{th}}$ percentile from i.i.d. 100 independent draws of $x$ from $\bar{p}$. Figs. 13b and 13d plot estimated upper bounds of $\mathbb{E}[L_{\bar{p},\epsilon}(\hat{X}_N)]$ (averaging over 100 independent draws from $q$) against flow length $N$. The choice and detailed derivation of the upper bounds of $L_{\bar{p},\epsilon}$ can be found in Appendix C.3

# D  Additional experimental details

All experiments were conducted on a machine with an AMD Ryzen 9 3900X and 32G of RAM.

## D.1  Model description

The two synthetic distributions tested in this experiment were

- the banana distribution [51]:

$$y = \begin{bmatrix} y_1 \\ y_2 \end{bmatrix} \sim \mathcal{N}\left(0, \begin{bmatrix} 100 & 0 \\ 0 & 1 \end{bmatrix}\right), \quad x = \begin{bmatrix} y_1 \\ y_2 + by_1^2 - 100b \end{bmatrix}, \quad b = 0.1;$$

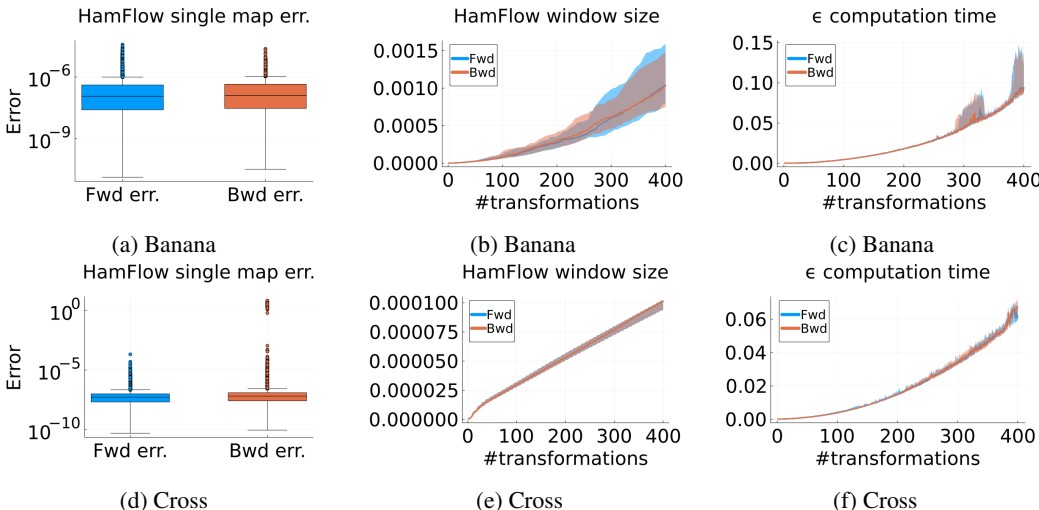

Figure 14: `HamFlow` forward (`fwd`) and backward (`bwd`) single transformation error $\delta$ on both synthetic examples (Figure (a) and (d)), shadowing window size $\epsilon$ over increasing flow length (Figure (b) and (e)), and shadowing window computation time (Figure (c) and (f)). Figure. (a) and (d) show the distribution of forward error $\left\{ \|F^n x_0 - \hat{F} \circ F^{n-1} x_0\| : n = 1, \ldots, N \right\}$ (`Fwd err.`) and backward error $\left\{ \|F^n x_0 - \hat{B} \circ F^{n+1} x_0\| : n = 1, \ldots, N \right\}$ (`Bwd err.`). These errors are computed along precise forward trajectories that originate at $x_0$, which is sampled 50 times independently and identically from the reference distribution $q_0$. $N$ here denotes the flow length for each example. Figure (b) and (e) show $\epsilon$ over increasing flow length. Figure (c) and (f) show the wall time of computing shadowing window over increasing flow length. The lines in Figure (b)–(c) and (e)–(f) indicate the median, and error regions indicate $25^{\text{th}}$ to $75^{\text{th}}$ percentile from 50 runs.

- a cross-shaped distribution: a Gaussian mixture of the form

$$x \sim \frac{1}{4}\mathcal{N}\left(\begin{bmatrix} 0 \\ 2 \end{bmatrix}, \begin{bmatrix} 0.15^2 & 0 \\ 0 & 1 \end{bmatrix}\right) + \frac{1}{4}\mathcal{N}\left(\begin{bmatrix} -2 \\ 0 \end{bmatrix}, \begin{bmatrix} 1 & 0 \\ 0 & 0.15^2 \end{bmatrix}\right)$$
$$+ \frac{1}{4}\mathcal{N}\left(\begin{bmatrix} 2 \\ 0 \end{bmatrix}, \begin{bmatrix} 1 & 0 \\ 0 & 0.15^2 \end{bmatrix}\right) + \frac{1}{4}\mathcal{N}\left(\begin{bmatrix} 0 \\ -2 \end{bmatrix}, \begin{bmatrix} 0.15^2 & 0 \\ 0 & 1 \end{bmatrix}\right);$$

The two real-data experiments are described below.

**Linear regression.** We consider a Bayesian linear regression problem where the model takes the form

$$\textbf{Lin. Reg.:} \; \beta \overset{\text{i.i.d.}}{\sim} \mathcal{N}(0, 1), \log \sigma^2 \overset{\text{i.i.d.}}{\sim} \mathcal{N}(0, 1), \quad y_j \mid \beta, \sigma^2 \overset{\text{indep}}{\sim} \mathcal{N}\left(x_j^T \beta, \sigma^2\right),$$

where $y_j$ is the response and $x_j \in \mathbb{R}^p$ is the feature vector for data point $j$. For this problem, we use the Oxford Parkinson's Disease Telemonitoring Dataset [52]. The original dataset is available at `http://archive.ics.uci.edu/dataset/189/parkinsons+telemonitoring`, composed of 16 biomedical voice measurements from 42 patients with early-stage Parkinson's disease. The goal is to use these voice measurements, as well as subject age and gender information, to predict the total UPDRS scores. We standardize all features and subsample the original dataset down to 500 data points. The posterior dimension of the linear regression inference problems is 21.

**Logistic regression.** We then consider a Bayesian hierarchical logistic regression:

$$\textbf{Logis. Reg.:} \; \alpha \sim \mathsf{Gam}(1, 0.01), \beta \mid \alpha \sim \mathcal{N}(0, \alpha^{-1} I), \quad y_j \mid \beta \overset{\text{indep}}{\sim} \mathsf{Bern}\left(\frac{1}{1 + e^{-x_j^T \beta}}\right),$$

We use a bank marketing dataset [53] downsampled to 400 data points. The original dataset is available at `https://archive.ics.uci.edu/ml/datasets/bank+marketing`. The goal is to use client information to predict subscription to a term deposit. We include 8 features from dataset: client age, marital status, balance, housing loan status, duration of last contact, number of contacts during campaign, number of days since last contact, and number of contacts before the current campaign. For each of the binary variables (marital status and housing loan status), all unknown entries are removed. All included features are standardized. The resulting posterior dimension of the logistic regression problem is 9.

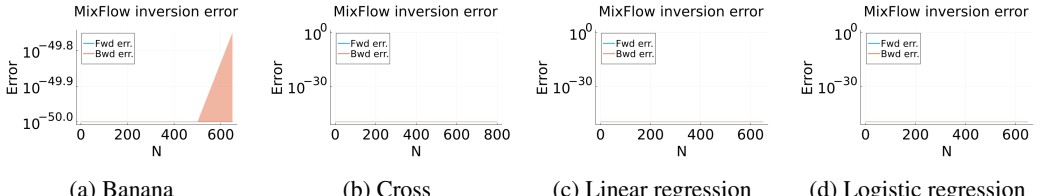

(a) Banana      (b) Cross      (c) Linear regression      (d) Logistic regression

Figure 15: MixFlow inversion error when using 2048-bit `BigFloat` representation. Verticle axis shows the reconstruction error $\|B^N \circ F^N(x_0) - x_0\|$ (`Fwd`) and $\|F^N \circ B^N(x_0) - x_0\|$ (`Bwd`) for the two synthetic examples (Figs. 15a and 15b) and the two real data examples (Figs. 15c and 15d). $F$ and $B$ are implemented using 2048-bit `BigFloat` representation, and $x_0$ is sampled from the reference distribution $q_0$. The lines indicate the median, and the upper and lower quantiles over 10 independent runs. It can be seen that in all four examples, the inversion error of MixFlow is ignorable when using 2048-bit computation.

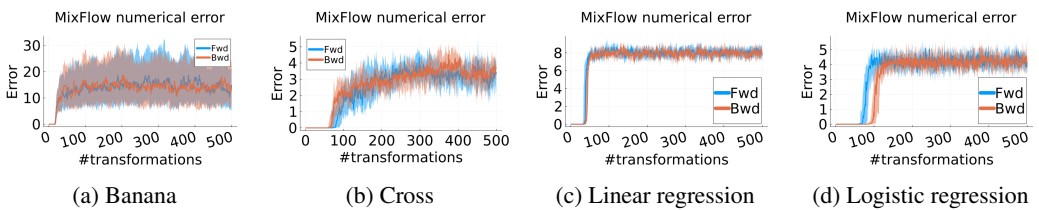

(a) Banana      (b) Cross      (c) Linear regression      (d) Logistic regression

Figure 16: MixFlow forward (`fwd`) and backward (`bwd`) orbit errors on both synthetic examples and real data examples. (a)—(d) show the median and upper/lower quartile forward error $\|F^k x - \hat{F}^k x\|$ and backward error $\|B^k x - \hat{B}^k x\|$ comparing $k$ transformations of the forward exact/approximate maps $F$, $\hat{F}$ or backward exact/approximate maps $B$, $\hat{B}$. Note that Figs. 1b, 1e, 1c and 1f respectively display the initial segments of (a)—(d) in a logarithmic scale, intended to provide a clearer illustration of the exponential growth of the error.

## D.2    MixFlow parameter settings

For all four examples, we use the uncorrected Hamiltonian MixFlow map $F$ composed of a discretized Hamiltonian dynamic simulation (via leapfrog integrator) followed by a sinusoidal momentum pseudorefreshment as used in Xu et al. [5] (see [5, Section 5, Appendices E.1 and E.2] for details). However, we use a Gaussian momentum distribution, which is standard in Hamiltonian Monte Carlo methods [54] as opposed to the Laplace momentum used previously in MixFlows [5]. In terms of the settings of leapfrog integrators for each target, we used 200 leapfrog steps of size 0.02 and 60 leapfrog steps of size 0.005 for the banana and cross target distributions respectively, and used 40 leapfrog steps of size 0.0006 and 50 leapfrog steps of size 0.002 for the linear regression and logistic regression examples, respectively. The reference distribution $q_0$ for each target is chosen to be the mean-field Gaussian approximation as used in [5].

As for the `NUTS` used for assessing the log-density evaluation quality on two real data examples, we use the Julia package `AdvancedHMC.jl` [55] with all default settings. `NUTS` is initialized with the learned mean of the mean-field Gaussian approximation $q_0$, and burns in the first 10,000 samples before collecting samples; the burn-in samples are used for adapting the hyperparameters of `NUTS` with target acceptance rate set to be 0.7.

## D.3    Additional MixFlow results

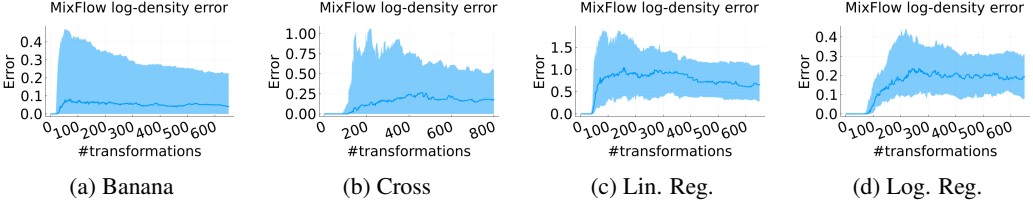

(a) Banana      (b) Cross      (c) Lin. Reg.      (d) Log. Reg.

Figure 17: MixFlow log-density evaluation error. For synthetic examples (Figs. 17a and 17b), we assessed densities at 100 evenly distributed points within the target region (Figs. 2a and 2b). For real data examples (Figs. 17c and 17d), we evaluated densities at the locations of 100 `NUTS` samples. The lines indicate the median, and error regions indicate $25^{\text{th}}$ to $75^{\text{th}}$ percentile from 100 evaluations on different positions.

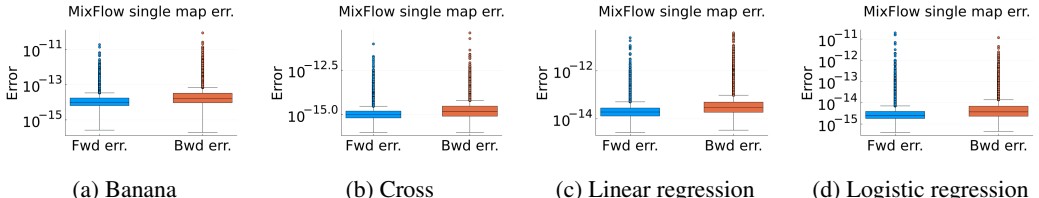

(a) Banana      (b) Cross      (c) Linear regression      (d) Logistic regression

Figure 18: MixFlow forward (`fwd`) and backward (`bwd`) single transformation error $\delta$ on both synthetic examples and real data examples. (a)—(d) show the distribution of forward error $\left\{ \|F^n x_0 - \hat{F} \circ F^{n-1} x_0\| : n = 1, \ldots, N \right\}$ (`Fwd err.`) and backward error $\left\{ \|F^n x_0 - \hat{B} \circ F^{n+1} x_0\| : n = 1, \ldots, N \right\}$ (`Bwd err.`). These errors are computed along precise forward trajectories that originate at $x_0$, which is sampled 50 times independently and identically from the reference distribution $q_0$. $N$ here denotes the flow length for each example.

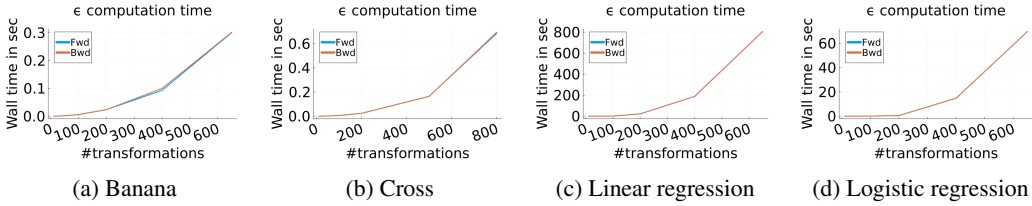

(a) Banana      (b) Cross      (c) Linear regression      (d) Logistic regression

Figure 19: MixFlow shadowing window computation time (wall time in second) for both the forward (`Fwd`) and backward (`Bwd`) orbits over increasing flow length. The lines indicate the median, and error regions indicate $25^{\text{th}}$ to $75^{\text{th}}$ percentile from 10 runs. Note that here the Monte Carlo error between different runs is so small that the error bar is too thin to be visualized.

