# OpenReview forum: "Embracing the chaos: analysis and diagnosis of numerical instability in variational flows"
_NeurIPS.cc/2023/Conference — NeurIPS 2023 poster_

### Official Review · Reviewer_e3zg · 2023-07-05

**Soundness:** 2 fair
**Presentation:** 3 good
**Contribution:** 2 fair
**Rating:** 5
**Confidence:** 3

**Summary:**

The authors present an analytical framework to quantify the effects of numerical errors on sampling, density and ELBO estimation in variational flows. The framework leverages the shadowing theory to derive error bounds based on the shadowing window sizes and the local Lipschitz constants of the dynamical operators. A procedure to verify whether the shadowing property holds in a given system is provided.

**Strengths:**

* [Significance] Numerical analysis is an important and under-studied aspect in flow-based models. This work takes a step forward in this direction and offers a plausible explanation for why fast accumulation of numerical errors do not necessarily lead to bad results.
* [Quality] The derivations of the numerical bounds seem to make sense intuitively - although I did not check the math in detail.
* [Clarity] The material is presented in a clear and logical manner. I find it easy to follow in general.
* [Novelty] There exists prior work that is based on shadowing theory but this work extends the analysis to the computation of density and ELBO estimations.


**Weaknesses:**

* My primary concern about this work is mainly on the usefulness of the presented framework in practice. All results are predicated on the basis that the shadowing property is already established. The approach presented in section 5 looks cumbersome and does not seem to scale particularly well.
* I do not find the numerical experiments to support the claims very well. They only demonstrate some qualitative traits - sampling, density and ELBO errors are much lower compared to trajectory errors, which coincides with the observation that the derived shadowing window sizes are relatively low. There is not much numerical evidence showing how tight the derived bounds are.


**Questions:**

* The analysis in section 4.3 is predicated on the basis that (ln 181) “when $F$ and $B$ are ergodic and measure-preserving for the target $\pi$”, which is a pretty strong assumption. Do you have ways to establish this?
* The horizontal axis in Figure 7 does not have the same range as those in Figures 4 to 6. Is this due to computational constraints, i.e. the proportionality to $N$?
* It appears that $\epsilon$ is still growing at the end while the sampling, density and ELBO errors all seem to have converged. Do you have any comments on the qualitative behaviors when extrapolating further? This would imply that the bounds eventually become very loose.


**Limitations:**

There is not much explicit discussion on the limitations. The computational cost for establishing and calculating the shadow window size is an obvious one.

---

> ### Author Rebuttal · Authors · 2023-08-09
>
> Thank you for reviewing our manuscript! Please see our point-to-point respoonse as follows.
>
> > My primary concern about this work is mainly on the usefulness of the
> > presented framework in practice. All results are predicated on the basis that
> > the shadowing property is already established.
>
> Please see our general response 5 for a discussion of assumptions and the scope of our work.
>
>
> > The approach presented in section 5 looks cumbersome and does not seem to scale particularly well.
>
> Your concern about the complexity and scalability of the approach presented in Section 5 is valid. However, we wish to highlight that the primary focus of this work is to build a theoretical framework to understand the numerical stability of variational flows, and to explain the counterintuitive phenomenon regarding numerical error in a rigorous manner. The present approach in Section 5 should be considered as a diagnostic tool that enables users to verify when our theory can be applied. It's not positioned as an algorithm that would be run each time variational flows are employed. Even so, as discussed in our general response 2, it is likely that a much more efficient algorithm for computing $\lambda$ exists.
>
> > "I do not find the numerical experiments to support the claims very well. They
> > only demonstrate some qualitative traits - sampling, density and ELBO errors
> > are much lower compared to trajectory errors, which coincides with the
> > observation that the derived shadowing window sizes are relatively low. There
> > is not much numerical evidence showing how tight the derived bounds are"
>
> Please see our general response 4 for discussion of experiments.
>
> > “It appears that $\epsilon$ is still growing at the end while the sampling, density
> and ELBO errors all seem to have converged. Do you have any comments on the
> qualitative behaviors when extrapolating further? This would imply that the
> bounds eventually become very loose.”
>
> This is indeed an interesting phenomenon; as mentioned in our general response 3, understanding how $\epsilon$ scales with $N$ in more generality is an important direction of future work. Indeed if $\epsilon$ continues to grow, we expect that eventually the bounds should become loose. Empirically, it seems to grow linearly with a gentle slope, and we suspect the range of $N$ for which these bounds are reasonable is large. But we cannot make a rigorous claim about this at this point.
>
> Nonetheless, we believe that our current results offer valuable insights and represent significant contributions to the community. To the best of our knowledge, our theory is the first among the variational flow literature that rigorously explains how numerical error influences the orbit evaluation and downstream tasks in different ways.
>
> > “The analysis in section 4.3 is predicated on the basis that (ln 181) “when
> F and B are ergodic and measure-preserving for the target”, which is a pretty
> strong assumption. Do you have ways to establish this?”
>
> To be clear, these assumptions are not required by all analysis in this section; they are just used to obtain stronger results when the dynamical system targets a particular distribution. Since ergodicity and pi-invariance are the two key conditions for convergence of MixFlows paper [5], and are also quite common in ergodic theory [Eisner15], we studied how shadowing would interact with those conditions in variational methods. Our intention was just to provide a new insight that a dynamical system targeting a particular distribution may behave even better numerically than a general system. Given that they are standard conditions, verifying them is beyond the scope of the present work.
>
> However, note that there are well-known measure-preserving systems often used in statistics (e.g. Hamiltonian flows, or an ODE satisfying the Liouville conditions [Neklyudov21]). Although one must discretize ODEs in practice, note that our numerically approximated map $\hat F$ can involve discretization and does not need to be measure-preserving. On the other hand, ensuring ergodicity is challenging in practice, as acknowledged in Section 4.3 of [5]. We agree that verifying ergodicity is a current limitation of MixFlows, but addressing this issue is beyond the scope of our work.
>
> [Eisner15] Eisner, T., Farkas, B., Haase, M., and Nagel, R. Operator
> Theoretic Aspects of Ergodic Theory. Graduate Texts in
> Mathematics. Springer, 2015.
>
> [Neklyudov21] Neklyudov, K., Bondesan, R., and Welling, M. Deterministic gibbs sampling via ordinary differential equations.
> arXiv:2106.10188, 2021.
>
>
>
> > “The horizontal axis in Figure 7 does not have the same range as those in
> > Figures 4 to 6. Is this due to computational constraints, i.e. the
> > proportionality to N?”
>
> This was just a minor oversight on our behalf. We originally chose the range of these plots to illustrate the typically linear (or slower) relationship between the shadowing window size and $N$. We can certainly make the range of $N$ match Figure 6 for the camera ready–thank you for pointing this out!
>
>
> > “There is not much explicit discussion on the limitations. The computational
> cost for establishing and calculating the shadow window size is an obvious
> one.”
>
> Please see general response 3 for a discussion of limitations, and general response 2 for a discussion of computational cost.

---

> > ### Comment · Reviewer_e3zg · 2023-08-21
> >
> > Thank you for your detailed response. I certainly agree that there is good value (in terms of novelty and significance) in the theoretical framework - that's why I gave a passing score to begin with. However, I do still believe more concrete evidence is needed to demonstrate the practical usefulness. Since these were not provided in the rebuttal, I remain with my previous score of 5.

---

> > > ### Author Response · Authors · 2023-08-21
> > > **Thank you**
> > >
> > > Thank you for your response and review comments -- we appreciate your insights!

---

### Official Review · Reviewer_d5rn · 2023-07-09

**Soundness:** 3 good
**Presentation:** 1 poor
**Contribution:** 3 good
**Rating:** 6
**Confidence:** 3

**Summary:**

This paper investigated the impact of accumulated numerical error on the sampling, density evaluation, and evidence lower bound (ELBO) estimation in variational flows. It demonstrated that the results produced by flows are not destroyed by the serious numerical instability. To explain this phenomenon, it leveraged shadowing theory to theoretically bound the error of sampling, density evaluation, and ELBO estimation. It also developed a diagnostic procedure that can be used to validate results produced by numerically unstable flows in practice.

**Strengths:**

This paper proposed an interesting question that why the accumulated numerical error does not influence the results of sampling, ELBO, etc. The technique is sound, and the analyses are solid. The derived theoretical results could support its claim.


**Weaknesses:**

The presentation could be improved, and the notations are hard to follow, see details below.


**Questions:**

1. What is the definition of "s_k" in theorem 4.1.
2. What is the range of mathcal{X}, is it belong to R or R^N or R\times[ 0,T]?
3. I cannot follow the information in Figure 1(b)(d), e.g., what do you want to show through the dots in these figures?

**Limitations:**

The limitations are not fully discussed in this paper, e.g., the theory only provide explanations while it does not inspire new algorithm for generative learning. It will narrow the impact of the theory that practitioners may ignore.

---

> ### Author Rebuttal · Authors · 2023-08-09
>
> Thank you for reviewing our manuscript! Please see our point-to-point respoonse as follows.
>
> > What is the definition of $s_k$ in theorem 4.1.
>
> "s" stands for "shadowing". $(s_k)_{k = 0}^N$ denotes an exact orbit starting at $s = s_0$ that "shadows" the numerical trajectory (as mentioned in line 132--133). Please refer to our definitions and illustrations of orbits, found between lines 117--126 and in Figure 4. We will make sure to clarify this explicitly in the camera ready version!
>
> > What is the range of $\mathcal{X}$, is it belong to $R$ or $R^N$ or $R\times[ 0,T]$?
>
> As outlined between lines 63–68, $\mathcal{X} \subseteq R^d$ represents the space on which the variational family $\\{q_\lambda: \lambda \in \Lambda\\}$ is supported.
>
> > I cannot follow the information in Figure 1(b)(d), e.g., what do you want to show through the dots in these figures?
>
> Figure 1(b)(d) serves to illustrate the impact of floating-point representation errors during flow computations. Ideally, in the absence of numerical errors, the computed orbit (blue scatters) would coincide perfectly with the exact orbit (red scatters). However, based on the figure, even for a relatively small $N$, there's a significant deviation of the numerical orbit from the exact one.
>
> > The limitations are not fully discussed in this paper, e.g., the theory only
> provide explanations while it does not inspire new algorithm for generative
> learning. It will narrow the impact of the theory that practitioners may
> ignore.
>
> Please see our general response 3 for a discussion of the limitations. Note that the purpose of this work is not to introduce a new methodology, but to provide a novel set of theoretical results that explain a counterintuitive, surprising phenomenon related to numerical errors in variational flows that we often see in practice.

---

### Official Review · Reviewer_sFEg · 2023-07-09

**Soundness:** 2 fair
**Presentation:** 3 good
**Contribution:** 2 fair
**Rating:** 5
**Confidence:** 3

**Summary:**

The paper "Embracing the chaos: analysis and diagnosis of numerical instability in variational flows" investigates the impact of numerical instability on the reliability of sampling, density evaluation, and evidence lower bound (ELBO) estimation in variational flows. The authors treat variational flows as dynamical systems and leverage shadowing theory to elucidate the abnormal behavior: common flows can exhibit a catastrophic accumulation of error, but surprisingly, results produced by flows are often accurate enough for applications despite the presence of serious numerical instability.

**Strengths:**

In terms of originality, the paper presents a new approach to understanding the behavior of variational flows in the presence of numerical instability. By treating variational flows as dynamical systems and leveraging shadowing theory, the authors provide theoretical guarantees on the error of sampling, density evaluation, and ELBO estimation. The paper also presents a diagnostic procedure to empirically validate results produced by numerically unstable flows in practice.

In terms of quality, the paper is well-written. The authors provide a clear motivation for their work and present their results in a coherent manner. The paper also includes several theorems, which adds to the quality of the work.

In terms of clarity, the paper is easy to follow and understand. The authors provide clear explanations of their methods and results, and the paper includes several figures and examples that help to illustrate the concepts presented.

In terms of significance, the paper makes mainly two contributions to the field of variational inference. The paper provides a theoretical framework to understand the behavior of variational flows in the presence of numerical instability, which is an important problem in the field. The paper also presents a diagnostic procedure to validate results produced by numerically unstable flows in practice.

**Weaknesses:**

One potential weakness of the paper "Embracing the chaos: analysis and diagnosis of numerical instability in variational flows" is that it focuses primarily on theoretical analysis and does not provide as much empirical validation of the proposed diagnostic procedure as would be desirable. While the authors do provide some empirical results to support their claims, it would be beneficial to see more extensive experiments that demonstrate the effectiveness of the proposed diagnostic procedure on more complex problems datasets like Cifar 10.

Another potential weakness of the paper is that the theoretical contribution with the concepts of dynamical systems and shadowing theory has an overlap with [1] , which is the main theoretical insight of the paper. Moreover, the theorems in ELBO esitimation and density estimation follows diretcly from the shadowing property.

Reference:
1. Paul Tupper. The relation between approximation and shaowing in molecular dynamics.

**Questions:**

1. Does the shaowing property holds for continuous normalizing flow (empirically or theoretically) ?

2. On theorem 5.1, the important constant $M$ depends on $x \in \mathcal{X}$, right?  If so, then we cannot derivative the shaowing property (which is a global property independent of $x$) from theorem 5.1.

**Limitations:**

The paper could benefit from a more detailed discussion of the limitations of the proposed approach.

---

> ### Author Rebuttal · Authors · 2023-08-09
>
> Thank you for reviewing our manuscript! Please see our point-to-point respoonse as follows.
> > “One potential weakness of the paper "Embracing the chaos: analysis and
> diagnosis of numerical instability in variational flows" is that it focuses
> primarily on theoretical analysis and does not provide as much empirical
> validation of the proposed diagnostic procedure as would be desirable. While
> the authors do provide some empirical results to support their claims, it
> would be beneficial to see more extensive experiments that demonstrate the
> effectiveness of the proposed diagnostic procedure on more complex problems
> datasets like Cifar 10.”
>
> Please see our general response 4.
>
> > “Another potential weakness of the paper is that the theoretical contribution
> > with the concepts of dynamical systems and shadowing theory has an overlap
> > with [1], which is the main theoretical insight of the paper. Moreover, the
> > theorems in ELBO esitimation and density estimation follows directly from the
> > shadowing property.”
>
> Please see our general response 1.
>
> > “Does the shadowing property holds for continuous normalizing flow (empirically or theoretically) ?”
>
> This is a good question, and is also something we are exploring now!  Generally speaking, shadowing theory also applies to numerical discretization of continuous flows such ODE (see e.g. the Finite shadowing theorem in [Coomes95]).  And the downstream tasks of continuous NFs or neural ODEs, e.g., sampling and density evaluation, indeed rely on the numerical discretization of the underlying ODE. Hence, it should be possible to theoretically analyze the error for continuous NFs via shadowing, but we leave a deeper investigation for future work.
>
> [Coomes95] Brian A. Coomes, Hüseyin Koçak, and Kenneth J. Palmer. Rigorous computational shadowing of orbits of ordinary differential equations. Numerische Mathematik, 1995.
>
>
> > “On theorem 5.1, the important constant M
>  depends on $x\in \mathcal{X}$, right? If so, then we cannot derive the shadowing property (which is a global property independent of $x$) from theorem 5.1.”
>
> This is a very good point! Indeed, Theorem 5.1 involves a value of $M$ that is valid for a single numerical trajectory. Then as long as we know something about the smoothness of $F_n$ locally around that numerical trajectory (e.g. 3rd derivative bound), we can obtain an upper bound on $M$ for that trajectory. But you are right that in order to prove that shadowing holds globally, we would need to bound $M$ for all starting states $x \in \mathcal{X}$. We can obtain such a bound if we make a global smoothness assumption on $F_n$ — e.g., a uniformly bounded 3rd derivative would suffice.
>
> We will certainly add a discussion of this in the camera ready. Thank you!
>
> On this note, a very interesting potential way to relax this is to investigate “probabilistic shadowing,” where there is a shadowing trajectory only with high probability under the initial distribution. We will add this to our discussion!

---

> > ### Author Response · Authors · 2023-08-11
> > **Correcting a minor error in our response**
> >
> > We hope to correct a minor error in our response to your last comment. To achieve a universal bound on M for all trajectories, we need a uniform bound on the 2nd derivative of the $F_n$, not the 3rd.

---

> > ### Comment · Reviewer_sFEg · 2023-08-18
> > **Response from the reviewer**
> >
> > I have no further questions. Hope the authors will address the assumptions of the theorems more carefully when updating the paper.

---

> > > ### Author Response · Authors · 2023-08-18
> > > **Thank you**
> > >
> > > Thank you again for your insightful feedback! We will indeed elaborate on the assumptions of Theorem 5.1 in the revised paper.

---

### Official Review · Reviewer_c44C · 2023-07-25

**Soundness:** 3 good
**Presentation:** 3 good
**Contribution:** 3 good
**Rating:** 7
**Confidence:** 3

**Summary:**

The paper discusses the (non)impact of numerical stability concerns when implementing variational flows, specifically, the robustness of sampling, density evaluation or ELBO computation against the chaotic behaviour of numerical implementations of variational flows.

The results are the following: while small perturbations of the input imply large perturbations of the output of variational flows (Section 3), the quality of samples, density evaluations, and ELBOs are rarely affected.
The paper explains this phenomenon with "shadowing":
More specifically (Section 4), under the assumption of ($\epsilon$, $\delta$)-shadowing, the errors of sampling, density evaluation, and ELBO computation behave like $\epsilon + d_\text{TV}(q_0, \xi_0)$, where $\xi_0$ is the initial condition that shadows $q_0$.
Whether or not a flow admits shadowing (and how $\epsilon$ relates to $\delta$) can be quantified by estimating (and bounding) the operator norm of the inverse of a specific operator that depends on the gradient(s) of the flow (Section 5). Numerical experiments show how the shadowing windows are minimal in typical applications of variational flows, even though numerical inaccuracy in implementing the flow is significant.



**Strengths:**

Analysing the numerical implementation of variational flows through the lens of dynamical systems is exciting because it logically explains a seemingly unlogical phenomenon. On top of that, the paper is very well written: Although some components are rather technical (e.g. shadowing), the manuscript remains easy to follow.


**Weaknesses:**

In general, I like the paper, but Section 5 ("Computation of the shadowing window size") leaves some open questions, especially in comparison to the finite-time shadowing Theorem in [33]. In my view, the following points should be addressed before publication:

1. The algorithm for computing $\lambda = \|A^{-1}\|$ is inaccurate: The implementation explained in Appendix B pretends that one can compute $\nabla F_k$ exactly, but since evaluating $F_k$ suffers from approximation errors, $\nabla F_k$ must as well. For context, compare Appendix B to Section 3.4 (especially, Equation 45) in [33] (reference in the paper). The inaccuracy is unlikely to be crucial but exists. It can (and should) be corrected. Or have I missed something?
2. Why does the paper estimate the operator norm the way it does? In other words, why is a call to the default `eigmin` function superior to the algorithm discussed in Section 3.4 in [33]? The paper mentions that the complexity of `eigmin` is $O(d^3N^3)$, whereas the tailored algorithm on p. 187 in Section 3.4 in [33] appears to cost $O(N d^3)$. Please discuss the numerical cost of estimating the shadowing windows in the experiments (for example, by providing estimates of the run-/wall time).
3. Could the authors please elaborate on how to estimate $M$? Why is estimating $M$ and $\delta$ problem-specific, and estimating $\lambda$ not (line 238)? If estimating $M$ is problem-specific, please discuss estimating $M$ for the problems in Section 6. (I have not found this information in Section 6 and Appendix D.)
4. Sections 5 and 6 seem to focus on the computation of the shadowing window size $\epsilon$. Does this mean all flows considered in this paper automatically have the shadowing property? Is this true for all variational flows (e.g., those discussed by Papamakarios et al.: neural spline, planar, radial, Hamiltonian, ...)? It would be helpful to discuss the existence of shadowing before computing the window size in the experiments (e.g. by estimating $2M\lambda^2\delta$).

As mentioned above, these points should be addressed before publication. But I expect all questions to be straightforward to answer, and I will increase my score once this is done.

**Questions:**

- This is not a question but more of a non-technical suggestion for the author(s). There have been discussions about potential ethical problems of the "Boston Housing" dataset: https://scikit-learn.org/1.0/modules/generated/sklearn.datasets.load_boston.html. If the corresponding experiment in the paper can be rerun with a different dataset, the longevity of this work may benefit.
- Out of curiosity: Why are the sampling bounds propositions but the ELBO and density bounds theorems?
- Please correct the complexity statements as commented in Appendix B.


**Limitations:**

As mentioned under "weaknesses" above, one potential weakness of the algorithm derived from Theorem 5.1 could be computational feasibility. From the manuscript, it is unclear how much computational power it takes to implement this algorithm. A reader will benefit from learning about this limitation if the algorithm is prohibitively costly for some applications.
Other than that, limitations seem to be discussed wherever relevant.

---

> ### Author Rebuttal · Authors · 2023-08-09
>
> Thank you for your review and comments! Before we address them, we would like to point out that [33] does not introduce the finite shadowing theorem—[17] does. We assume that when you refer to [33] in your review, you mean [17]. Hopefully we did not misunderstand!
>
> [17]: Brian Coomes, Hüseyin Koçak, and Kenneth Palmer. Shadowing in discrete dynamical systems. In Six lectures on dynamical systems, pages 163–211. World Scientific, 1996.
>
> [33]: Ivan Dokmanić and Rémi Gribonval. Beyond moore-penrose part i: generalized inverses that minimize matrix norms. arXiv:1706.08349, 2017.
>
> > The algorithm for ...
>
> This is an excellent point! Initially, we felt detailing the errors in computing $\lambda$ was too nuanced. However, we now agree with your view.  We plan to provide a more comprehensive description of computing $\lambda$, including handling the small numerical errors that occur there, in the final text.
>
> The fix is very straightforward, as you suggest – we use Eq. (45) in [17]. And it does not affect the results much: indeed if we compute terms in $\nabla F_k(x)$ with a floating-point error $\delta$, this introduces $O(\sqrt{N}\delta)$ Frobenius norm error into our representation of $A$, which then translates to an upper bound on $\lambda$ of roughly
> $$(1- \delta||\tilde{A}^{-1}||)^{-1}||\tilde{A}^{-1}|| = (1 - O(\sqrt{N}\delta) \tilde\lambda)^{-1} \tilde\lambda.$$
> Here $\tilde{A}$ denotes the digital computed $A$ and $\tilde\lambda := ||\tilde{A}^{-1}||$.
>
> Substituting values from our experiments, this amounts to an inconsequential relative error of around $(1- 10^{-9})^{-1} \approx 1$ on our minimum eigenvalues $\lambda$.
>
> > why does the ...
>
> This is also a great question!  We didn't claim `eigmin` outperforms the algorithm in Section 3.4 of [17]   (originally in [Coomes95] for general dimensions, so for the rest of the response we refer to [Coomes95]). You're correct that [Coomes95] offers better scaling than a direct call to `eigmin`.
>
> However, the method in [Coomes95] operates under the heuristic that a dynamical system with shadowing is hyperbolic, or nearly so. Specifically, the 'hyperbolic splitting' discussed in P.413 Appendix B of [Coomes95] (i.e., the choice of $\ell$), and the 'hyperbolic threshold' choice in P. 415 Appendix B of [Coomes95] (i.e., the choice of $p$), assumes that the dynamics is hyperbolic or close to it, which are impractical for our variational flows (our dynamics are general time-inhomogeneous systems). Also, as discussed in Appendix B (line 488--492), this method can potentially lead to a significant overestimation of the shadowing window when the dynamics are not (nearly) hyperbolic.
>
> Therefore, we prefer to use a procedure that is simpler and more widely applicable, at the cost of more computation. We found in our experiments that using `eigmin` took just a few seconds in most cases (see our general responses), which was acceptable. We will include the above discussion about the method proposed in [17] in the revised manuscript (likely in the appendix).
>
> [Coomes95] Brian A. Coomes, Hüseyin Koçak, and Kenneth J. Palmer. Rigorous computational shadowing of orbits of ordinary differential equations. Numerische Mathematik, 69:401–421, 1995.
>
> > Could the authors ...
>
> Estimating $M$ can indeed be problem-specific because $M$, by its definition (line 232), requires knowledge of the local Lipschitz smoothness of each flow layer evaluated at the computed orbit. This will necessitate different analyses for distinct flows. A universal approach might involve knowing a uniform upper bound on the 3rd derivative of $F_n$ (see response to Reviewer sEFg). Likewise, determining $\delta$ demands analyzing the single step error for the given flow.
>
> As for computing $\lambda$: we can use standard automatic differentiation tools to differentiate $F_n$ and compute $A$ (up to floating point error). At that point, we just need to compute a eigenspectrum for a symmetric blockwise tri-diagonal matrix. Both computations can be done in a generic way without model-specific derivations.
>
> > Sections 5 and 6 ...
>
> We appreciate your suggestion, and will include discussion about the existence of shadowing in the revised manuscript! It is not true that all variational flows exhibit the shadowing property. Our Theorem 5.1 states a sufficient condition, specifically, $M\lambda \epsilon \leq 1$ for the shadowing property to exist. The practical verification of the shadowing property requires an estimation of the one-step digital computing error $\delta$, the local smoothness constant of the flow $M$, and $\lambda$.
>
> Our manuscript describes the computation of $\lambda$ (via an eigenvalue problem for a symmetric blockwise tri-diagonal matrix) and the estimation of $\delta$ (Figure 10). These suffice to compute the shadowing window size. Estimating M can be more difficult, especially for complex flows. While one could analyze the local Lipschitz smoothness of each flow layer, handling the $\sup$ in the definition of M can be difficult.  One option is to bound  the 3rd derivative of $F_n$ to upper bound the $\sup$ analytically. However, given the range of results already in our manuscript, we leave this analysis for model-specific applications in future work.
>
> > This is not a question ...
>
> Thanks for pointing this out! We were not aware of this concern. Yes, we will re-run our experiments for the camera ready with another dataset.
>
> > Out of curiosity ...
>
> The sampling bounds were slightly easier to prove compared to the theorems. We can switch them all to “theorem” for consistency’s sake in the camera ready.
>
> > Please correct ...
>
> Will do!
>
> > As mentioned under "weaknesses" ...
>
> Thanks for your suggestion! We will provide more details regarding computing the shadowing window, with a thorough complexity analysis in the camera ready.  Please also see our general response 2 for a detailed response.

---

> > ### Comment · Reviewer_c44C · 2023-08-11
> >
> > Thank you for your reply.
> >
> > I am very sorry for mixing up the references. Indeed, I mean reference [17] -- I have no idea why I mentioned [33] instead.
> >
> >
> > **Regarding $\lambda$:** I agree with everything you said; please include the explanations in the revision.
> >
> >
> > **Regarding `eigmin`:** I concur that a simple, out-of-the-box solution like calling `eigmin` is a good starting point and may be sufficient for most applications. But I think a reader of your paper should know that a more efficient (albeit more complicated) method exists.
> >
> >
> > **Regarding $M$ and sections 5 and 6:**
> >
> > So would you agree with my summary that although it is difficult to determine whether a flow has the shadowing property (because bounding $M$ is difficult), computing the size of the shadowing window is straightforward (because $\lambda$ and $\delta$ are readily available)?
> > In other words, we can always compute the sizes of hypothetical shadowing windows but cannot guarantee that these windows exist.
> >
> > I find this very interesting. However, since -- strictly speaking -- it somewhat limits the helpfulness of Theorem 5.1 in practical applications, this phenomenon strengthens my wish to see more of a discussion of estimating (or at least bounding) $M$. If this is too much work (it may well be), then I would appreciate you at least explaining how one would approach this problem and why it is difficult.
> >
> > What do you think?

---

> > > ### Author Response · Authors · 2023-08-11
> > > **Response to further comments**
> > >
> > > Thank you for your continued response!
> > >
> > > > Regarding $\lambda$: I agree with everything you said; please include the explanations in the revision.
> > >
> > > Yes! We will certainly include the explanation in camera ready.
> > >
> > > > Regarding `eigmin`:  I concur that a simple, out-of-the-box solution like calling eigmin is a good starting point and may be sufficient for most applications. But I think a reader of your paper should know that a more efficient (albeit more complicated) method exists.
> > >
> > > We agree! We will provide an expanded discussion on the method provided in [Coomes95] (the current manuscript contains a brief discussion in Appendix B, line 488–492) and will add a discussion of potentially more scalable algorithms that leverages the sparsity of $AA^T$ as described in the general response 2.
> > >
> > > > Regarding M and sections 5 and 6: So would you agree with my summary that although it is difficult to determine whether a flow has the shadowing property (because bounding M  is difficult), computing the size of the shadowing window is straightforward (because $\lambda$  and $\delta$ are readily available)?
> > >
> > > You are totally correct.
> > >
> > > > If this is too much work (it may well be), then I would appreciate you at least explaining how one would approach this problem and why it is difficult.
> > >
> > > We appreciate your constructive comment about estimating $M$. To clarify, the challenge of demonstrating shadowing is as follows:
> > > 1. To demonstrate shadowing for *one* numerical trajectory, you need the supremum of $\nabla^2F_n$ locally around each numerical trajectory point (i.e., $M$ as stated in Theorem 5.1 of our manuscript).
> > > 2. To demonstrate shadowing for *all* numerical trajectories (i.e., our Definition 4.1), you need to bound the supremum of $\nabla^2F_n$ globally.
> > >
> > > As in our response to sFEg and to your previous comment, we can derive bounds as follows. To obtain a bound on $M$ for a particular trajectory, we just need $\nabla^2F_n(\hat{x}_n)$ for each numerical orbit point (e.g., via automatic differentiation), as well as a bound on the norm of the 3rd derivative of $F_n$. For example, if $\|\nabla^3F_n\| \leq B$ uniformly, then
> > > $$
> > > M \leq \max_n \{ \epsilon B + \|\nabla^2F_n(\hat{x}_n)\|\}.
> > > $$
> > > For a global bound on M, we need a universal bound on the 2nd derivative of $F_n$. For example, if $\|\nabla^2 F_n\|\leq B$ uniformly, then $M \leq B$. Note that in both cases, the uniform bound on the derivative requires an in-depth analysis tailored to the specific flow.
> > >
> > > It is also possible to estimate the value of $M$ for an individual trajectory by computationally maximizing $\|\nabla^2 F_n\|$ in the $\epsilon$-ball around the numerical orbit points. For example, one could sample points randomly within the $\epsilon$-ball, compute $\|\nabla^2 F_n\|$ for each, and estimate the supremum using those values. However, this approach is very computationally expensive in general, and we would not recommend it generally.
> > >
> > > However, even without direct analysis of $M$, we can judge how large $M$ would have to be to violate the shadowing condition. In particular, for our experiments, for the shadowing property to be violated (i.e., $M>(\epsilon\lambda)^{-1}.$),  $M$ would need an order of magnitude greater than $10^{8}$. This is an unlikely scenario for flows that are reasonably smooth.

---

> > > > ### Comment · Reviewer_c44C · 2023-08-14
> > > >
> > > > That makes sense - thank you for clarifying. I consider my questions to be answered and update my score accordingly. Thank you!

---

> > > > > ### Author Response · Authors · 2023-08-18
> > > > > **Thank you**
> > > > >
> > > > > Thank you for engaging with us and for your constructive feedback! We'll make sure to address and expand upon the discussed points in the final version.

---

### Official Review · Reviewer_MECE · 2023-07-29

**Soundness:** 2 fair
**Presentation:** 3 good
**Contribution:** 3 good
**Rating:** 6
**Confidence:** 3

**Summary:**

The paper investigates the impact of numerical instability on the reliability of sampling, density evaluation, and evidence lower bound (ELBO) estimation in variational flows. It demonstrates through empirical examples that numerical instability can lead to deviations in the flow map affecting sampling, density, and ELBO computations. However, the paper also finds that despite numerical instability, results from flows can still be accurate enough for practical applications. The paper treats variational flows as dynamical systems and uses shadowing theory to provide theoretical guarantees on the error of downstream tasks. It also develops a diagnostic procedure to validate results produced by numerically unstable flows in practice. The paper concludes by validating its theory and diagnostic procedure on MixFlow with both synthetic and real data examples.

**Strengths:**

* The paper addresses an important and practical problem in the context of variational flows, which are widely used in generative modeling and probabilistic inference.
* The empirical examples and experimental validation provide concrete evidence for the theoretical claims made in the paper.
* The use of shadowing theory from dynamical systems provides a rigorous framework to understand the behavior of numerically unstable flows.
* The diagnostic procedure developed in the paper could be valuable for practitioners using variational flows, allowing them to assess the reliability of their results.

**Weaknesses:**

* Due to rather strong assumptions, the paper's claims may not be applicable in all scenarios. The experiments focus only on a few datasets as well as on a specific type of flow (MixFlow). Thus, they may not fully capture the behavior of other types of variational flows or other datasets, which limits the generalizability of the findings.

* It should be explained in more detail what novelty the current submission offers compared to the work of [23] (except for an extension to the evaluation of densities and ELBO estimations).

**Questions:**

* It appears that the exponential scaling in the density evaluation is circumvented by looking at *log*-densities?

* Can one motivate the assumptions (in particular, $\xi_0=q_0$) for the ELBO result?

**Limitations:**

Details on limitations are lacking.

---

> ### Author Rebuttal · Authors · 2023-08-09
>
> Thank you for reviewing our manuscript! Please see our point-to-point respoonse as follows.
>
> > “Due to rather strong assumptions, the paper's claims may not be applicable in all scenarios.”
>
> Please see our general response 5 for a comprehensive discussion of the assumptions.
>
> > “The experiments focus only on a few datasets as well as on a specific type
> > of flow (MixFlow). Thus, they may not fully capture the behavior of other
> > types of variational flows or other datasets, which limits the
> > generalizability of the findings.”
>
> Thank you for the comment! Although the focus of this paper is theoretical, we are happy to include more experiments in the camera ready; see our general response 4 regarding this comment.
>
> > “It should be explained in more detail what novelty the current submission
> > offers compared to the work of [23] (except for an extension to the
> > evaluation of densities and ELBO estimations).”
>
> Please see our general response 1 for a detailed comparison to [23].
>
> > “It appears that the exponential scaling in the density evaluation is circumvented by looking at log-densities?”
>
> This is a very good point! We did not adequately comment on this in the original submission, and will do so in the camera ready. There are two cases to consider here – absolute error and relative error.
>
> If one cares about the absolute error of the density, then our result suffices without much additional work. Densities tend to be bounded, while log-densities become unbounded in the tails. Generally, if two distributions are pointwise close in log-density, they will be pointwise close in density as well. Things become slightly trickier for unbounded densities, but we will stick with bounded here for simplicity.
>
> If one cares about the relative error of the density, then indeed our result suggests that exponential growth in the relative error is possible. But in this case, this is again a major improvement on what you would expect; recall that the empirical phenomena we observe is that the *trajectories themselves* diverge exponentially. So if the trajectories diverge exponentially quickly, then the relative error could grow *doubly-exponentially*.
>
> As a rough example, consider the ratio of two normals: $\mathcal{N}(0,1)$, and $\mathcal{N}(x, 1)$. Then if we consider the map $x \to \exp(x)$ — a toy model for an exponential divergence in pushforward error—we obtain density ratios like $\exp(-\theta^2 + (\exp(x) - \theta)^2)$, which grow doubly exponentially in $x$.
>
> > “Can one motivate the assumptions (in particular, $\xi_0 = q_0$) for the ELBO result?”
>
> This is a good point, and we will clarify this for the revised manuscript. The assumption $\xi_0 = q_0$ has been made in past work [23], and was made here primarily for convenience of the analysis (lines 175–177). But it is a reasonable assumption for the following reason. We know that $\xi_0$ is close to $q_0$ due to shadowing.
> And $\xi_0$ is indeed an implicit function of $q_0$; it’s a fixed point of a twice differentiable function involving the whole trajectory starting at $q_0$, see page 176 of [17] (we do not recommend to solve the fixed point equation in practice). The implicit function theorem yields that $\xi_0$ is a differentiable function of the numerical orbit (whose initial distribution is $q_0$). So, the relationship between $q_0$ and $\xi_0$ is not totally pathological. Assuming that $q_0 = \xi_0$ provides simplicity without compromising the insights of the result.
>
> > Details on limitations are lacking.
>
> Please see our general response 3 for a comprehensive discussion of limitations.

---

> > ### Comment · Reviewer_MECE · 2023-08-16
> > **acknowledgment**
> >
> > Thank you for the clarifications. After reading the general response as well as the responses to other reviews, I updated my score under the assumption that both the theoretical explanations, as well as further numerical experiments, will be included in the final version.

---

> > > ### Author Response · Authors · 2023-08-18
> > > **Thank you**
> > >
> > > Thank you again for reviewing our work! We will enhance our theoretical discussions and expand the numerical experiments in the revised manuscript.

---

### Author Rebuttal · Authors · 2023-08-09

# General response to the reviewers

We thank the reviewers for their valuable feedback. In this response, we address shared comments. Specific responses to each reviewer will follow separately.

## 1. Comparison to [23] (MECE, sFEg)

Our work is not an extension or competitor to the work of [23]; both focus on different objectives. While [23] establishes a relationship between shadowing and the weak distance between inexact and exact trajectories in dynamical systems, our work provides a theoretical basis for the counterintuitive behavior of numerically-implemented variational flows observed in practice. The overlap is minimal; our Proposition 4.2 echoes the main theorem in [23] with some differences (e.g., Levy-Prokhorov vs. bounded Lipschitz distance, our work does not place assumptions on the initial state distribution). We introduce a substantial set of new results in Theorems 4.3, 4.4, 4.6, 4.7, and Proposition 4.5 dedicated to density evaluation, ELBO estimation, and similar and strengthened results for MixFlows. These theorems are not consequences of  [23], employing distinct proof techniques. Our work offers significant contributions even for readers familiar with [23].

We'll elaborate on the relationship with [23] in the camera ready.

## 2. Computational cost of the shadowing window (c44C, e3zg).

In our experiments, this was a fairly minor computational cost. For a flow length of N=500, `eigmin` took just a few seconds. Given that this was acceptable in practice, we didn’t code a specialized method. In the camera ready, we will report the computation time for estimating the shadowing window size.

Even so, we suspect that $\lambda$ can be computed more efficiently by utilizing the sparsity of $AA^T$: specifically, $AA^T$ is a symmetric positive definite block-tridiagonal matrix with bandwidth $d$ (line 476--477 in Appendix B) and so has $O(Nd^2)$ entries. We are currently investigating more efficient methods for calculating $\lambda$ along these lines, e.g., the inverse power method [Schatzman02, Chapter 13.3.3], or tridiagonalization via Lanczos iterations [Lanczos] followed by divide-and-conquer algorithms [Coakley13].

[Schatzman02] Michelle Schatzman. Numerical analysis: a mathematical introduction. Oxford University Press, USA, 2002.

[Lanczos] C. Lanczos: An Iteration Method for the Solution of the Eigenvalue Problem of Linear Differential and Integral Operators, J. Res. Nat. Bur. Stand. 49, 255 (1950)

[Coakley13] Ed S. Coakley and Vladimir Rokhlin. A fast divide-and-conquer algorithm for computing the spectra of real symmetric tridiagonal matrices. Applied and Computational Harmonic Analysis, 2013.



## 3. Discussion of the limitations (MECE, sFEg, d5rn, e3zg)

We agree there was not enough discussion of limitations in the original submission; given the extra page in the camera ready, we are happy to include a thorough discussion of the limitations of our work, based on the following key points:

- Our error bounds mainly address downstream tasks post-training; we do not provide results regarding how numerical instability influences the training process itself.
- While our theory is centered around error bounds that are proportional to the shadowing window size $\epsilon$, we’ve observed that $\epsilon$ can grow with $N$. This suggests further study on its theoretical growth rate; we provided an initial discussion of this in the Appendix, but more work needs to be done here.
- Our Theorem 5.1 verifies shadowing for a particular trajectory, but needs a global smoothness condition on $F_n$ for global shadowing. To help address this, our current analysis may be extended from the global shadowing property to a weaker version that only holds with high probability.  See our response to Reviewer sFEg.
- Our theoretical analysis is focused on standard variational flow methods, and is not adapted to recent architectures like continuous normalizing flows or neural ODEs.
- In our experiments, we employed a basic method to calculate the minimum eigenvalue of $AA^T$. Given its sparsity, a deeper exploration into more efficient techniques is merited.
- Finally, we could include a broader range of variational flows (e.g., RealNVP, GLOW, and Hamiltonian variational flow) in our experiments.

## 4. Lack of numerical experiments (MECE, sFEg, e3zg)

We agree that more experiments that cover other types of variational flow would be a nice addition to the paper. We will plan to include results for more types of flow in the camera ready.

However, we’d like to emphasize that more experimental results are not central to the contributions of this particular work. The key contributions of this work are theoretical: we provide results that use the shadowing property to explain why the error of variational flows in statistical applications remains controlled, despite the catastrophic error growth in their numerical trajectories (Figure 1(b)(d)). Our experiments are designed just to provide empirical illustration of the theory and investigate the typical shadowing window size.

## 5. Usefulness of the theory (shadowing is a strong assumption) (MECE, e3zg)

We respectfully disagree that finite shadowing is an overly strong assumption. Indeed, we believe this is precisely the right assumption to explain the counterintuitive error growth phenomena in many common dynamical systems, even though this assumption of course does not hold for all systems. Further, we provide a sufficient condition to verify shadowing in practice via Theorem 5.1, which makes the strength of the assumption a priori less of an issue.

As a side note: even beyond finite shadowing, the *infinite* shadowing property is generic for dynamical systems (informally, there is an infinite shadowing dynamical system “arbitrarily close to” any reasonable dynamical system; see submission Appendix C). This suggests finite shadowing, as a weaker property, should be similarly pervasive.

---

### Decision · Program_Chairs · 2023-09-21

**Decision:**

Accept (poster)

**Comment:**

In this study, the authors delve into the repercussions of numerical instability on the reliability of various processes, including sampling, density evaluation, and evidence lower bound (ELBO) estimation, within the context of variational flows. The investigation begins with empirical examinations and shows that a commonly employed flows can encounter a detrimental accumulation of errors. This manifests in the form of significant deviations between the numerical flow map and its exact counterpart, leading to repercussions on sampling. Similarly, the numerical inverse flow map fails to accurately recover the initial input, thereby influencing the accuracy of density and ELBO computations. They develop theory to illuminate this behavior through theoretical guarantees concerning the errors in sampling, density evaluation, and ELBO estimation.

The reviewers thought the paper addresses an interesting and important problem, the experiment provide good evidence, and use of shadowing theory is interesting and the diagnostic procedure can be of value. They did have some concerns about the assumptions and novelty and overlap with some papers. The authors response seems to have alleviated these concerns and the reviews all recommend acceptance. I concur.